# Molecular Mechanism and Regulation of Autophagy and Its Potential Role in Epilepsy

**DOI:** 10.3390/cells11172621

**Published:** 2022-08-23

**Authors:** Hanxiao Zhu, Wei Wang, Yun Li

**Affiliations:** 1Clinical Medical School, Dali University, Dali 671000, China; 2Department of Neurology, The First Affiliated Hospital of Dali University, Dali 671000, China; 3Kunming College of Life Science, University of Chinese Academy of Sciences, Kunming 650000, China

**Keywords:** autophagy, autophagosome, epilepsy, synapse, mTOR, pathological mechanism

## Abstract

Autophagy is an evolutionally conserved degradation mechanism for maintaining cell homeostasis whereby cytoplasmic components are wrapped in autophagosomes and subsequently delivered to lysosomes for degradation. This process requires the concerted actions of multiple autophagy-related proteins and accessory regulators. In neurons, autophagy is dynamically regulated in different compartments including soma, axons, and dendrites. It determines the turnover of selected materials in a spatiotemporal control manner, which facilitates the formation of specialized neuronal functions. It is not surprising, therefore, that dysfunctional autophagy occurs in epilepsy, mainly caused by an imbalance between excitation and inhibition in the brain. In recent years, much attention has been focused on how autophagy may cause the development of epilepsy. In this article, we overview the historical landmarks and distinct types of autophagy, recent progress in the core machinery and regulation of autophagy, and biological roles of autophagy in homeostatic maintenance of neuronal structures and functions, with a particular focus on synaptic plasticity. We also discuss the relevance of autophagy mechanisms to the pathophysiology of epileptogenesis.

## 1. Introduction

Autophagy is a highly conserved catabolic process that ensures cellular homeostasis through the digestion and recycling of misfolded proteins, damaged organelles, and invasive pathogens [1]. Among three mechanistically distinct types, macroautophagy is the major and multistep process that involves the formation of double-membrane autophagosome, which is different from microautophagy and chaperone-mediated autophagy (CMA). In macroautophagy, selected cytoplasmic components are sequestrated in autophagosomes via a series of dynamic membrane events, such as appearances, expansion, and closure of phagophore, and maturation, trafficking, and fusions of the autophagosome, as well as release and reuse of degradation products [2]. This process virtually occurs in almost all mammalian cells, including rodents and primates, implicating in cell quality control, development and differentiation, activity-dependent function, survival, aging, and other cell activities [3].

Unlike other cell types, neurons are post-mitotic cells and have an unusual, compartmentalized morphology, such as axons, dendrites, and synaptic connections, which play a key role in neurotransmission, neural circuits, synaptic plasticity, and other neuronal physiology [4]. These unique characteristics determine that neurons require high demands on autophagy to maintain homeostasis and ensure survival over many decades [5]. Although neuronal autophagy shares basal machinery with non-neuronal cells, some neuron-specific proteins including endophilin A and synaptojanin1 are involved in the autophagy mechanisms for maintaining synaptic morphology and functions, particularly at axon terminals [6]. A growing body of evidence suggests that activity-dependent synaptic plasticity, the critical basis of brain adaptation, relies on neuronal autophagy [7,8]. Along with this prominent role in neuronal homeostasis, autophagy is also essential for the occurrence and progression of neurological and neurodegenerative diseases, such as epilepsy, which is characterized by spontaneously synchronized abnormal neuronal firing activity [9]. Recent studies have shown a close link between epilepsy and autophagy [10]. Pathological changes in epilepsy manifestations such as synaptic structural and functional changes, neuronal excitatory and inhibitory imbalances, and abnormal connections in neural circuits are also regulated by autophagy [10,11]. However, the potential role of autophagy in epilepsy has been relatively unexplored. 

In this review, we describe the historical landmarks, distinct types, and core machinery of autophagy in the cell and the interconnection between neuronal subcellular compartments and autophagy. We will then discuss the physiology regulations of autophagy in synapse plasticity and explore the recent progress in neuronal autophagy in epilepsy.

## 2. The Historical Landmarks of Autophagy

Soon after the observation of the existence of cytoplasmic organelles within membrane-limited vacuoles by Clark in 1957, Ashford and Poter first found that after glucagon stimulation, the components from other organelles such as mitochondria were detected in some lysosomes that moved to the center of the hepatic cell, and appeared to be progressive decomposition [12,13]. Unfortunately, they mistakenly believed that the above phenomenon was the process of lysosome formation and that lysosomes were not organelles located in the cytoplasm like mitochondria. In 1963, Hruban et al. reported the ultrastructure of focal cytoplasmic degradation including three consecutive steps from cytoplasmic fusion to lysosome formation. They proposed that this process is not only induced by the injury stage but also in the physiological stage of organelle disposal, cytoplasmic component reuse, and cell differentiation [14]. Based on Hurban’s discoveries, de Duve’s laboratory named this phenomenon “autophagy” and appreciated that lysosomes play a vital role by fusion with prelysosomes containing sequestered cytoplasm in the cellular autophagy triggered in the rat liver by glucagon [15,16,17]. Remarkably, Clarke proposed a new death type for dying cells, autophagic cell death (ACD), distinct from apoptosis or caspase-independent cell death [18]. Although the importance of autophagy and lysosomes for maintaining cell homeostasis had been noted by researchers at the time, it was mainly regarded as a system for non-specific cleaning up and scavenging debris. 

Degradation of cytoplasmic components was considered to be one of the key functions of the vacuoles of yeast cells, but it remained largely unresolved how cellular components are delivered to the vacuole. Many decades later, Ohsumi and colleagues identified the extensive autophagic degradation of cytosolic components in the vacuoles of yeast cells and named spherical bodies “autophagic bodies”, which contained cytoplasmic ribosomes, rough endoplasmic reticulum (ER), mitochondria, lipid granules, and glycogen granules [19]. Accumulation of autophagic bodies in the vacuoles of yeast cells was induced not only by nutrient-deficient conditions but also by disruption of the *PRB1* gene (*PRB1* gene encodes proteinase B associated with protein degradation with the vacuoles) or deficiency of multiple proteinases. Simultaneously, they isolated a mutagenized proteinase-deficient strain, apg1, which is defective in the accumulation of the autophagic bodies in the vacuoles, and confirmed that at least 15 *APG* genes are involved in autophagy, which is indispensable for protein degradation via autophagic bodies in the vacuoles of yeast cells under starvation conditions [20]. This result was arguably the most important historical event in the field of autophagy (Figure 1).

Following the introduction of genetics, autophagy-related genes had also been identified by other researchers independently. For example, Thumm et al. identified *AUT* genes (found to be identical to *APG* genes later) from autophagocytosis mutants of *Saccharomyces cerevisiae* [21]; Sakai et al. identified *GSA* genes during a search for underlying molecular involved in peroxisomal degradation [22]. Due to confusion about the names of the field, these autophagy-related genes and proteins had been unified as “*ATG*” and “Atg” in 2003, respectively, which stand for “autophagy-related” [23]. Strikingly, the Ohsumi laboratory cloned the *ATG1* gene for the first time and explored the genetic interaction between ATG1 and ATG13 in yeast [24], and reported a well-conserved Atg12-Atg5 conjugation system that uses a ubiquitination-like conjugation model and is essential for autophagy in yeast and humans [25,26]. Meanwhile, *beclin 1*, a mammalian homolog of yeast *Atg6/Vps30*, was cloned by Liang et al. in 1999, and decreased expression of which may contribute to the development of human breast carcinoma and other malignancies [27]; Microtubule-associated protein 1 light chain 3 (LC3), a mammalian homolog of yeast Atg8p, was found to be specifically associated with autophagosome membranes after processing and has since been used as a marker of autophagy [28].

To determine how the isolation membranes are formed in autophagy, the Yoshimori laboratory showed that the Atg12-Atg5 conjugation system plays a critical role in isolation membrane development using green fluorescent protein-tagged Atg5-deficient mouse embryonic stem cells [29]. This is the first study on the molecular mechanism of autophagosome formation in mammals. Simultaneously, the signaling pathways in which distinct classes of phosphatidylinositol 3’-kinases (PI3K) control macroautophagy of HT-29 cells in opposite directions were discovered by Petiot et al. [30]; Rapamycin, a specific inhibitor of the mammalian target of rapamycin (mTOR), was found to induce autophagy and attenuate polyglutamine protein accumulation in cell models of Huntington’s disease [31,32]. This method of rapamycin-inducing autophagy has been widely used in the field of autophagy. Furthermore, the development of tools for monitoring autophagy by GFP-LC3 transgenic mice and useful marker proteins has accelerated the expansion of autophagy research across many fields such as neuropsychiatric disease, structural biology, developmental biology, aging biology, cell metabolism, inheritance and evolution, cancer, drug addiction, immunity, infection, and inflammation, to name a few [33,34,35,36,37,38,39]. In addition, the findings of organelle-specific autophagy reflect a corresponding multiplicity of intracellular events, such as pexophagy, mitophagy, reticulophagy, nucleophagy, lysophagy, and ribophagy [16,40,41,42,43,44,45,46,47,48].

Recent work has demonstrated that the pre-autophagosomal structure (PAS) in yeast is a liquid-like condensate of Atg protein complex and that inhibiting phase separation impairs PAS formation in vivo, indicating that phase separation plays an active, key role in autophagy, whereby it organizes the autophagy machinery at the PAS formation [49]. However, accumulating data has suggested that the ATG genes have functions in pathways other than autophagy and that some ATG elements may not be absolutely necessary for autophagy [50,51]. Even autophagy regulation shows obvious differences among different creatures [52]. Therefore, elucidating the meticulous structure and function of each component of distinct stages of autophagy will help us fully comprehend the autophagic system’s diversity.

## 3. The Distinct Types of Autophagy

Since its initial description, the autophagy process has mainly been referred to induction, sequestration, transportation, and degradation of unwarranted components of the cell [53]. Although the exact membrane resources (such as mitochondria, ER, Golgi, and plasma membranes) for autophagosomes remain a contentious question in the field, lysosomal digestion of the substrates has been widely recognized to date [2,54,55,56]. After the initiation of the process, different pathways for delivery of cargo to the lysosomes distinguish the three categories within autophagy in cells: CMA, microautophagy, and macroautophagy [57,58,59,60,61].

CMA was the first discovered process by which intracellular proteins are selectively degraded [62]. It involves the discrimination of degradation targets, the assembly of chaperone complexes, and other sophisticated mechanisms, which unfold and translocate the selected substrate across the lysosomal membrane, to then be digested by lysosomal hydrolases [61]. Specifically, target components for CMA were determined by a canonical KFERQ-like motif, which contains one of the negatively charged D or E residues, one or two of the positive residues K and R, one or two of the hydrophobic residues F, I, L, or V, and generally flanked by glutamine on one of the sides. On binding to the KFERQ-like motif, targeted proteins were docked to a LAMP2A-containing lysosomal membrane complex for degradation via the cytosolic chaperone HSC70 and cochaperones, such as HSP70-HSP90 organizing protein (HOP), the carboxyl terminus of HSC70-interacting protein (CHIP), and heat shock protein 40 (HSP40) [63]. By degrading specific target substrates promptly, CMA is involved in a variety of cellular activities.

Microautophagy is a constitutive process by which cytosolic components are directly engulfed by the invagination of endosomal, lysosomes, or vacuole membrane (in yeast) [64]. Due to the inability to detect an invagination-like process and no conserved function of yeast-microautophagy genes in mammals, the study of microautophagy has been least well elucidated.

Macroautophagy is an intricate and sequential process in which intracellular proteins and organelles are sequestered within the autophagosome, and then finally delivered to the lysosome for degradation [65,66]. During this process, phagophore, a double-membrane sequestering structure from several plausible sources of the cellular membrane, emerges first and becomes specialized as a PAS [67]. In the cooperation of two conserved ubiquitin-like conjugation systems, a portion of cytoplasm, including invasive pathogens, superfluous and damaged organelles, and cytosolic protein aggregates, is enveloped by elongation of PAS, ultimately forming a new vesicle, the autophagosome. Following delivery and fusion with the lysosome, the cargo including contents and its inner membrane is degraded by lysosomal hydrolases, and the breakdown products such as amino acids are released back to the cytoplasm for recycling the macromolecular constituents to reuse as building blocks or a source of energy under unfavorable conditions (Figure 2) [68]. In addition, ATG proteins are involved in each of these steps and mechanistically regulate the autophagic flux along autophagosome and autophagolysosome biogenesis [25,26,49]. Due to the highest efficient degradative capacity, macroautophagy is a predominant and prevalent degradative mechanism in various aspects of cellular activities.

Moreover, some researchers divided autophagy into non-selective and selective autophagy according to distinct nutritional statuses. For example, selective autophagy included mitochondrial autophagy (Mitophagy), endoplasmic reticulum autophagy, peroxisome autophagy under nutrient-rich conditions, etc. [69]. Notably, autophagy in this review is macroautophagy. 

## 4. The Molecular Machinery of Autophagy

Autophagy induction is generally determined by the cellular state. In nutrient-deficient conditions, AMP-activated protein kinase (AMPK) is activated by high AMP levels, thereby increasing the functions of the tuberous sclerosis complex (TSC1) and TSC2, which in turn inhibits the activity of the GTP-binding protein Rheb and phosphorylation of the mammalian target of rapamycin (mTOR) kinase [70,71]. Once mTOR is inactivated, it promotes the unc-51-like autophagy activating kinase 1 (ULK1) autophosphorylation, initiating PAS membrane assembly for phagophores biogenesis [72]. Regardless of the indirect mechanism, rapamycin, low amino acid levels, and energy stores also suppress mTOR phosphorylation. Furthermore, TSC1 and TSC2 are alternatively activated by the class I PI3K/Akt/protein kinase B (PKB) signal cascade, which is suppressed by the upstream growth factors or insulin, subsequently facilitating ULK1 phosphorylation by the Rheb/ mTOR pathway (Figure 3) [30]. In this situation, these modifications finally bring about the accumulation of phosphorylated ATG protein ULK1. 

ULK1, a protein with serine/threonine kinase activity, is the sole kinase of the core autophagy component in mammals [73]. Active ULK1 phosphorylates ATG13 and FIP200 (or RB1CC1), and recruits Atg101 (or C12orf44), a stabilizer of ATG13 and FIP200, forming the stable ULK1-ATG13-FIP200-Atg101 complex in a hierarchical manner (Figure 3) [74,75]. ATG13 is the regulatory subunit in this complex and identifies highly curved lipid membranes. In addition, the interaction between ULK1 and ATG13 is stabilized by the scaffold protein FIP200 [76]. In entire ULK1 complex-activated conditions, it initiates the assembly of phagophore membranes and facilitates the transitions into the PAS structures. Although the ULK1 complex is believed to be the core molecular machine of autophagy in the induction phase, the innate character of the PAS about its membrane origins and protein composition is not known.

Vesicle nucleation of autophagy is mainly regulated by the ATG9 cycling system and BECN1-ATG14L-VPS15-VPS34 complex (Figure 4) [66,77,78]. ATG9, which is a transmembrane protein consisting of six transmembrane domains, is phosphorylated by active ULK1 in PAS, subsequently forming a cycling system with ATG2 and WIPI1/2 (WIPI: WD-repeat protein interacting with phosphoinositides) [79]. Activated ATG9 shuttles between new growing PAS and membrane donors, such as the Golgi complex, endoplasmic reticulum, mitochondria, and other resources, and binds to the lipid-binding protein ATG13 in the ULK1 complex, thus directing membranes to the nascent PAS [80]. As for ATG2, it not only brings lipids to the growing PAS membrane as a possible lipid transporter but also interacts with ATG9 and WIPI1/2, in which WIPI1 and WIPI2 may oligomerize and interact with membranes for autophagy flux [81]. 

Simultaneously, the WIPI also links VPS34 (or PIK3C3) for phagophores expansion [82]. VPS34, the type III phosphatidylinositol-3 kinase (PI3K) that can be inhibited by 3-methyl adenine (3-MA), forms the class III phosphoinositide 3-kinase complexes with VPS15 (or PIK3R4), ATG14L and BECN1 (or VPS30) [83]. The activation and recruitment of the class III PI3K complex are dependent on the phosphorylation of BECN1 via active ULK1 [84]. It has been suggested that the class III PI3K complex binds to the C-terminus of the Atg101 HORMA domain in the PAS membrane, and generates the phosphatidylinositol-3 phosphate (PI3P), an essential component of the vesicle nucleation of phagophore during biogenesis [85]. The generation of PI3P facilitates the recruitment of PI3P effectors and PI3P-binding proteins, thus involving in “isolation” membrane formation, curvature shaping, and phagophores expansion [86]. In addition, ATG14 is a critical component for the proper recruitment and localization of the class III PI3K nucleation complex as well as the creation of an “isolation” membrane (Figure 4) [87]. 

Remarkably, BECN1, namely Beclin1, includes three key functional domains, the central coiled-coil domain (CCD), the Bcl2-homology3 (BH3) binding site located at the N-terminus of the protein framework, and the evolutionarily conserved and C-terminus domain, playing a core role in coordinating the formation of the class PI3K complex and the process of membrane transport [85,88]. More research has revealed that BECN1 is specifically required for autophagy vesicle nucleation, inflammation response, cell apoptosis, longevity and other cellular activities, and is considered as the central link of these signal pathways [89,90]. For instance, using BECN1-induced autophagy, the oncolytic viruses significantly increased BCR/ABL protein expression, and significantly improved the pathology of chronic myeloid leukemia, suggesting that BECN1 may offer a new target for clinical implications [91]. Meanwhile, the BECN1-related complex contains at least three different complexes, with the catalytic core region BECN1-VPS34 (Figure 4) [92]. BECN1-ATG14L-VPS34 complex, the classical autophagy pathway in vesicle nucleation, interacts with the phagocytic membrane of the endoplasmic reticulum, and is stimulated and inhibited by the binding of AMBRA1 and Bcl-2/Bcl-xL, respectively [66,93]; BECN1-VPS15-Vps34-UVRAG complex is the endocytic pathway of the autophagosome, and constitutively displaces AMBRA1 and ATG14 components with UVRAG and its regulatory element SH3GLB1, leading to the direct phagocytic effect and the accelerated maturation of autophagosomes [94]; The third complex is an inhibitory complex, in which the regulator SH3GLB1 is switched to KIAA0226/Rubicon. Rubicon can inhibit the UVRAG and VPS34 activities, finally playing a negative role in PAS membrane expansion and autophagosome maturation [95,96]. Additionally, Rubicon and ATG14L competitively interact with the central coiled-coil domain of BECN1, thereby maintaining the neutral state between endocytosis and autophagy [97].

Elongation of the phagophores membrane is main mediated by two ubiquitin-like (Ubl) conjugation systems, namely the microtubule-associated protein 1A/1B-light chain (LC3) Ubl conjugation and the ATG12 Ubl conjugation (Figure 5) [98]. Initially, under the effects of the E1-like ubiquitin ligase ATG7 and E2-like ubiquitin ligase ATG10, the carboxy-terminal Ubl structure of Ubl protein ATG12 conjugates to the lysine of the ATG5, forming the ATG12-ATG5 conjugate [99]. This conjugate increases the affinity for ATG16L1, which safeguards the ATG12-ATG5 complex against ubiquitin-like modification and protease degradation by binding to free ATG5, promoting the complicated assembly of the ATG12-ATG5-ATG16L1 system [100]. In this system, ATG16L1 plays the role of dimerization protein through binding to two ATG12-ATG5 conjugates and constitutes ATG12-ATG5-ATG16L1 homodimers [101]. ATG12 is one of two Ubl proteins in autophagy and it directs the ATG12 Ubl conjugation system to the outer membrane of the phagophores by an auxiliary connection [98]. Moreover, two hydrophobic residues in the ATG12 ubiquitin-fold region are necessary for the ATG12 Ubl conjugation system. The mutation at Y149 reduces the catalytic capability of ATG10 and severely impedes the conjugate formation of the ATG12-ATG5 complex, which is not affected by the F154 mutation [102]. In response to the mutation at F154, the LC3 II-phosphatidylethanolamine (PE) conjugation is significantly stopped, failing vesicle elongation [103].

In the LC3 Ubl conjugation system, the carboxy-terminal of LC3 is cleaved by ATG4 protease to form LC3-I, and then it is transferred into LC3-II formation via the action of E1-like ligase ATG7 and E2-like ligase ATG3 [104]. Subsequently, the ATG12 Ubl conjugation system interacts with LC3-II in a manner similar to the E3-like enzyme and facilitates the generation of LC3 II-PE conjugation through the amine headgroup binding of LC3-II and PE. LC3 II-PE conjugation relies on the ATG12 component to accomplish membrane positioning with the expanded phagophores membrane [105]. Lipidated LC3 not only discriminates against specific cargo through the LC3 interaction region domain but also participates in the elongation and closure of the PAS membranes and further autophagosome maturation (Figure 5) [104]. Moreover, LC3 is one of two subfamilies of yeast ATG8 homologs in mammals, the other being GABARAP [106]. Previous studies have shown that the LC3 proteins act at the stage of vesicle elongation, whereas the GABARAP functions at a later stage of autophagosome maturation [107]. Notably, LC3 or GABARAP proteins interact with a series of adaptor proteins, such as p62/sequestosome 1 (SQSTM1) (hereafter p62) and calcium-binding and coiled-coil domain-containing protein 2 (also called NDP52), facilitating autophagosome formation around specific cargoes [108,109,110]. p62 is one of the most well-characterized of these adaptors and is a multidomain protein that modulates distinct signaling pathways through different binding partners. Prominent p62 domains encompass the N-terminus Phox/Bem1p (PB1), the LC3-interacting region (LIR), the tumor necrosis factor receptor-associated factor 6 (TRAF6)-binding sequence (TBS), the ZZ-type zinc finger (ZZ) domain, the Kelch-like ECH-associated protein 1 (Keap1)-interacting region (KIR), and a C-terminal ubiquitin associated (UBA) domain. p62 interacts directly with LC3 and GABARAP protein family members through its LIR domain, and binds directly to selected cargoes through its C-terminal UBA domain [111]. Moreover, the PB1 domain-mediated homopolymerization of p62 results in its co-aggregation with the selected cargo, which promotes the close interaction between the p62-binging cargo and lipidated LC3 proteins (or GABARAP proteins) at the phagophore [112]. Not only are these interactions indispensable for delivery of selected substrates to autophagosomes, but they also potentially play a key role in the formation of autophagosomes themselves. Additionally, p62 degradation also relies on autophagy, that is, the level of p62 accumulates in response to the inhibition of autophagy, and the level of p62 decreases in response to the induction of autophagy [108,109]. Apart from this, accessory ATG proteins are related to the ultimate steps of autophagosome biogenesis, but their concrete roles are not explicitly known yet.

After the elongation, the cup-shaped PAS membrane encloses a portion of the cytoplasm or damaged organelles through the endosomal sorting complex required for transport (ESCRT) mechanism, finally forming the autophagosome [113]. Nascent autophagosomes are randomly dispersed throughout the cytoplasm, and recruit the molecular switch Rab GTPases to the autophagosome membrane, promoting the interaction between the autophagosome and dynein (Figure 6) [114]. Generally, Rab GTPases are critical regulators of eukaryotic membrane trafficking, such as Rab2A, Rab5, Rab7, and Drosophila ADP-ribosylation factor (Arf)-like 8 (Arl8) (key players in autophagy), and can hydrolyze GTP to GDP as well as bind to target membranes on GTP binding, thereby serving as the molecular identity of the target membrane (lysosomes, autophagosomes, late endosomes, and Golgi-derived vesicles) or molecular switches [115]. For instance, autophagosomes recruit Rab7 during maturation in a Monensin sensitivity protein 1-Caffeine, calcium, and zinc 1 (Mon1-Ccz1) dependent manner, and also play a key role during endolysosome formation or endosome maturation [113,116,117]. These proteins bind mainly to effectors required to drive the fusion of autophagosome-lysosome in mammals, such as cytoskeleton-associated motor proteins, tethering factors, and adaptors [118]. As is known to all, autophagosome and lysosome transport depends on the microtubular system and related motor proteins. The minus-end directed dynein-dynactin motor complex drives the autophagosome to the perinuclear region, whereas plus-end directed kinesins move late endosomes/lysosomes to the cell periphery, promoting each other’s successful encounters [119,120]. Rab GTPases including Rab7 bind to several motor adaptor proteins, such as FYVE and coiled-coil domain-containing 1 (FYCO1, mediate outward transport), Rab-interacting lysosomal protein (RILP, mediate inward transport), and oxysterol-binding protein-related protein 1L (ORP1L, mediate inward transport), pleckstrin homology and RUN domain containing M1 (PLEKHM1), and BORC complex [113,121,122,123]. These effectors can link lysosomes or autophagosomes to cytoskeleton-associated motor proteins (such as the light chain of kinesin 1 (KLC2), Kinesin-3, and kinesin-13), so they can mediate both plus-end and/or minus-end transport of these vesicles or autophagosome [124,125]. Of note, motor adaptor proteins have been shown to directly bind LC3 or GABARAP proteins to promote dynein-mediated autophagosome transport [126].

Through dynein-dependent mechanisms, autophagosomes and lysosomes are transported along the intracellular microtubules toward the perinuclear region where the autophagolysosome formation is mediated via the soluble N-ethylmaleimide sensitive fusion protein attachment protein receptor (SNAREs), the membrane-tethering proteins and other fusion-related protein families [113,127]. Specifically, autophagosome-lysosome (vacuole in yeast) fusion requires tethering the lysosome and autophagosome together, which is usually mediated by multisubunit membrane-tethering complexes or large coiled-coil proteins [128]. These proteins usually bind to GTP-loaded Rab GTPases and/or vesicle membranes of autophagosomes and late endosomes/lysosomes, and collaborate with Sec1/Munc-18 (SM) family proteins to promote SNAREs complex assembly and zippering [129]. The most well-studied tethering protein in autophagosome-lysosome fusion is the heterohexameric homotypic fusion and vacuole protein sorting (HOPS) complex, which comprises six subunits: the two Rab7 (Ypt7 in yeast) binding subunits (Vps41 and Vps39) and the 4-subunit class C core [130]. Accumulated pieces of evidence have indicated that HOPS is critical for nearly all lysosome-related fusion events, such as autophagosome-lysosome fusion, late endosome-lysosome fusion, and secretory granule-lysosome fusion during crinophagy and lysosome-related organelle biogenesis [127,131,132,133]. HOPS can directly (or indirectly) bind to Rab7 (through adaptors Plekhm1 and RILP), Rab2 (or RAB2A), and Arl8, and be recruited to lysosomes through Arl8 and the BORC complex and to autophagosomes through Rab2 or STX17. In addition, HOPS also acts as an SM protein via subunit Vps33A [134,135,136,137]. Besides HOPS, the tethering of autophagosomes with lysosomes is also meditated by additional proteins, such as BRUCE and RUN and FYVE domain-containing protein 4 (RUFY4), ectopic P-granules 5 (EPG-5), ATG14, Golgi reassembly stacking protein 55 (GRASP55), and tectonin beta-propeller repeat containing 1 (TECPR1) [138,139,140,141]. Remarkably, some tether proteins have also been found to bind to the autophagosome PI3P and LC3/GABARAP family proteins [142].

Tethered proteins cross-link the lysosomal or late endosomal membranes to autophagosome membranes, whereby SNARE complexes carry out autophagosome-lysosome fusion [143]. Approximately forty soluble SNARE proteins are essential for intracellular membrane fusion and they are classified into vesicular/v-SNAREs (R-SNAREs) and target/t-SNAREs (Q-SNARE-s). Q-SNAREs are usually grouped into Qa-, Qb-, and Qc-SNARE-s, and some are Qbc-SNARE-s [144,145]. Autophagosome-lysosome fusion is driven by a trans-SNARE complex, which consists of three different Q-SNAREs (Qa, Qb, and Qc-SNAREs) anchored in a vesicle membrane and an R-SNARE anchored to the membrane of another vesicle [144,146]. Interestingly, these SNARE proteins are assembled into multiple SNARE complexes, which are not conserved from yeast to mammalian cells [143]. For instance, the autophagosomal R-SNARE Ykt6, the vacuolar Qa-SNARE vacuole morphology3 (Vam3), and the Qc-SNARE Vam7 and the Qb-SNARE Vps10 Interacting1 (Vti1) form a trans-SNARE complex that promotes the membrane fusion of autophagosomes with the vacuole in yeast [147,148,149,150]. Conversely, autophagosome-lysosome fusion in mammals is mainly mediated by two different SNARE complexes: the autophagosomal Qa-SNARE Syntaxin17 (STX17), the lysosomal/late endosomal R SNARE Vamp7 or Vamp8, and two Qbc-SNARE SNAP-29; and the autophagosomal R SNARE YKT6, the lysosomal/late endosomal Qa SNARE STX7, and two Qbc-SNARE SNAP-29 [151,152,153,154,155]. Following the fusion with lysosomes or late endosomes, autophagosome components are digested by activating lysosomal hydrolase, and the degradation of macromolecules is transported back into the cytoplasm for recycling [156]. The ultimate destination of autophagolysosomes is to become residual bodies (Figure 6) [157].

Notably, the protein LC3-II and p62 levels have been widely applied to the assessment of autophagy flux in vivo and in vitro [158]. Autophagic flux is an entire intracellular process including acquisition, packaging, trafficking, degradation, and release of unwanted or damaged substrates in the lysosomal system. Monitoring autophagic flux is critically important for dissecting the biology of autophagy, renewing the understanding of how autophagy regulates physiological and pathological conditions, and screening for autophagy-modulating drugs or other molecular compounds [159]. To date, autophagic flux is usually measured by the comparison of the LC3-II level or the autophagosome number between samples or individuals with and without lysosomal inhibitors (chloroquine, bafilomycin A1, or other protease inhibitors) using fluorescence visualization strategies (such as red fluorescent protein, or GFP or mCherry) [160]. In parallel with monitoring LC3-II, the localization and level of p62 are used to reflect autophagic activity. In addition, several detection methods are currently available, such as flow cytometry, western blots, pulse-chase measurement of half-life, immunofluorescence (subcellular localization pattern) and electron microscopy, and live cell imaging with fluorescence protein tags (pH-sensitive or not) [161]. However, accurate measurement of autophagic flux is still challenging.

## 5. The Regulation of Autophagy in Neurons

Neurons are a class of post-mitotic, polarized, hypermetabolic, long-lived, and highly specialized cells that were initially elucidated by Cajal in the early 19th century. Unlike somatic cells, neurons have a unique architecture with three subcellular compartments: the soma, dendrites, and axons, which are critical to their highly specialized functions (such as signaling transmission and synapse plasticity) (Figure 7). Axons usually grow from the axon hillock and extend multiple branches and elaborate terminal arbors from growth cones, which connect widely divergent synaptic targets in brain regions, finally forming complex neural circuits and topographic maps [162]. In this network, synapses are the fundamental units of communication between two neurons and are established by the dendrites of one neuron and axons of the ascending projection neuron.

Although the neuronal architectures and characteristics (such as hypermetabolic, long-lived, and highly specialized) rely on cell determination, autophagy is required for the maintenance of these properties and functions. Firstly, the neuron is a long-lived and post-mitotic cell, whereby it has no ability to dilute the accumulated effects of damaged proteins, organelles, and themselves through mitosis. Autophagy can dominate the turnover of RNA, lipids, proteins, and organelles, which maintain neuronal survival throughout the lifetime of most organisms [163,164]. Secondly, the neuron is a polarized cell and has highly polarized structures, such as axons and dendrites. Autophagy plays a key role in these aspects (such as complex neural circuits and topographic maps). For instance, during early growth and active developmental refinement of axons and dendrites (such as orphan neurites are pruned), autophagy regulates new protein synthesis, protein degradation, and neurite elimination (axon or dendrite pruning), which is important for the formation of specific neuronal structures [163,165,166]. Thirdly, the elimination and establishment of synapses (synapse plasticity, and neuronal activity) are highly dynamically regulated in response to activity-dependent plasticity or Hebb’s synaptic plasticity that involves cytoskeletal dynamic mechanisms and multiple intracellular signaling cascades [167]. This process needs a lot of molecules and/or proteins (such as neurotransmitters, RNA, and lipids) that are synthesized, delivered, and degraded for their dynamic functions in synaptic growth and synaptic activity. These are also regulated by autophagy (see below) [168]. Thus, we can say that autophagy is critical to maintaining the specialized functions of the neuron.

In addition, neurons must function continuously for as long as possible in a starved state, which is exceedingly different from other cell types. Brain neurons perform the functions of receiving, transmitting, integrating, processing, and outputting information, actively managing the general and cognitive activities of the organism. These processes place high energy and metabolic demand on neurons, especially in axons and dendrites located far away from the soma. Previous studies have suggested that neurons can use several different energy sources optimally, receive additional nutrients and neurotrophin support from glial cells, and benefit from the hypothalamic regulation of peripheral nutrient supplies [169]. Autophagy can also regulate nutrient supply under extreme starvation or diabetes.

Notably, the activity-dependent refinement of synapses (including the neuronal circuit change) occurs during the development and learning stages, and shares features with diseases such as epilepsy. For instance, the structural remodeling of mossy fiber synapses and the formation of aberrant synaptic contacts (such as synapse plasticity) in the dentate gyrus are common features in experimental models of epilepsy and human TLE [170,171,172,173]. In addition, amino acid metabolism (such as GABA) may be critical for epilepsy due to its role as the neurotransmitter (synapse transmission) and energy source [11]. Epileptic seizures increase the energy requirements of metabolically already compromised neurons, establishing a vicious cycle that results in worsening energy failure and neuronal death [174]. Changes in action potentials and ion channels are also related to the epileptogenesis of both genetic and acquired epilepsies [175]. Thus, changes in these physiological states may lead to the occurrence and development of epilepsy pathology (see below).

### 5.1. Autophagy in Axons

Axons have a powerful capability to regulate growth, which controls the axonal extension to the accurate sites of descending projection neurons and the timely termination of unwanted growth, ensuring the successful establishment of functional neural networks [162]. Simultaneously, synaptic vesicles, fusing with the presynaptic membrane of the axon terminals, trigger the release of neurotransmitters into the synaptic cleft (or Postsynaptic density, PSD), and evoke the activity of postsynaptic neurons by binding to the postsynaptic membrane receptors. Then, axon integrity (including membranes recycling, protein composition, synaptic connection, etc.) and synaptic stability are also maintained over time [176]. Moreover, experience-dependent synaptic plasticity, such as long-term depression (LTD) and long-term potentiation (LTP), results in long-lasting structural changes via synaptic pruning, a process of depleting highly intracellular ATP [177]. All these processes in axons require autophagic recycling of damaged or no longer needed contents to repurpose materials for structural alteration and growth. Thus, autophagy is a critical mechanism for maintaining synaptic functions, particularly in presynaptic compartments of the axon.

Autophagy in axons is initially identified in cultured peripheral neurons, in which large autophagosomes are trafficked retrogradely toward the soma and fused with lysosomes, suggesting that the formation of autophagy appears constitutively in distal axons [178]. Subsequently, autophagosome biogenesis at the axonal tip was corroborated in the cultured dorsal root ganglion (DRG) neurons as well as the embryonic cortical and hippocampal neurons [179]. These findings have consistently occurred in multiple model organisms, such as *Caenorhabditis elegans*, yeast, worms, Drosophila, Zebrafish, rats, mouse, humans, and other mammalians, demonstrating that the biogenesis and trafficking of axonal autophagosome in vivo are evolutionarily conserved [180,181,182,183]. Conditional ablation of autophagy proteins ULK/UNC-51, ATG5, ATG7, ATG9A, and FIP200 indicated that impairing autophagy resulted in abnormal axon growth and morphology, selective accumulation of tubular ER, and increased neurotransmitter release, finally inducing axon degeneration and behavioral deficits in vivo [184,185,186,187]. In cultured motor neurons, Pleckstrin homology containing family member 5 (Plekhg5, the guanine exchange factor for Rab26 GTPase) gene inactivation decreased axon outgrowth and increased accumulation of synaptic vesicles at the axon tip, whereas constitutively active Rab26 GTPase mitigates functional deficits in axon growth and autophagy [188].

A recent study has shown that Rab26 and ATG16L regulate neurotransmitter release through autophagy. Rab26 localizes on synaptic vesicles and is preferentially oligomerized to its GDP-bound form, which binds to ATG16L1 (Figure 8) [189]. This complex links synaptic vesicle clusters to the autophagy pathway and may assist in the migration of synaptic vesicles to nearby active sites, involving the regulation of presynaptic autophagy and neuronal activity [190,191]. Collectively, accumulated evidence implicated that axonal autophagy shares core machinery with non-neuronal cells.

Apart from the general autophagy mechanisms, several neuron-specific proteins are involved in autophagy regulation at axon terminals. Endophilin A, a synapse-enriched adaptor that can be phosphorylated by the kinase LRRK2 at serine 75 of the BAR domain, generates zones of highly curved membranes [192]. These membranes serve as a docking platform to recruit E2-like ligase ATG3, facilitating autophagosome biogenesis (Figure 8a) [193]. The functional disturbance of phosphorylated Endophilin A accelerates the accumulation of clathrin-coated vesicles, and impedes synaptic transmission, thus resulting in severe activity-dependent neurodegenerations and seizures [194]. Furthermore, Endophilin A also interacts with GTPase dynamin and synaptojanin 1 by its SH3 domain for autophagy [195]. Dynamin is a core component of the endocytic machinery and interacts directly or indirectly with multiple accessory proteins to coordinate vesicle membrane fission, promoting the constriction of coated pits (Figure 8) [196]. Synaptojanin1 is a lipid phosphoinositide phosphatase that is localized to axon terminals and contains two different phosphatase domains, SAC1 and 5-phosphatase, playing a critical role in synaptic vesicle trafficking [197]. The SAC1 hydrolyses phosphoinositide phosphate PI(3)P, PI(4)P, and PI(3,5)P2, while the 5-phosphatase mainly targets the PI(4,5)P2 as an enzyme substrate [191,192]. The genetic disruption of synaptojanin1 enzymatic domains has confirmed that the 5-phosphatase domain reduces the membrane-binding affinity of the clathrin adaptors and uncoat nascent vesicles, rather than the SAC1 domain [198]. Interestingly, recent work has suggested that the SAC1 domain can demolish the WIPI2/Atg18a, which recognizes and binds the protein PI(3)P/PI(3,5)P2, and is essential for autophagosome biogenesis at presynaptic terminals, not the 5-phosphatase domain [191,199]. The mutations of synaptojanin1 R258Q caused the accumulation of Atg18a-positive structures at newly formed synaptic autophagosomes, and impaired the autophagosome maturation, finally inducing the loss of dopaminergic neurons in flies and humans [199,200]. In addition, the E3-like ubiquitin ligase FBXO32/atrogin-1 tabulates vesicle membranes and localizes to clathrin-coated regions for autophagosome formation in a manner similar to endophilinA [201].

Several studies have shown that autophagy deficiency induced by impairing positive regulators (such as EPG5, AP4, TECPR2, Alfy, VAMP7, PTPRσ, and other accessory proteins) exhibits different results in axon growth and neurological disease, accompanied by confirmations of gain or loss of function [202,203,204,205,206,207]. Some cause neuroaxonal dystrophy and hypoplasia, while others promote axonal growth. For instance, AP-4 is a member of the heterotetrameric adaptor protein (AP) family involved in protein sorting in the endomembrane system of eukaryotic cells (including neurons) [208]. δ2 glutamate receptor (δ2R) and amino-3-hydroxy-5-methyl-4-isoxazolepropionic acid (AMPA)-type glutamate receptors (AMPAR) are also sorted by AP-4 as cargos [209,210]. The AP-4 ε (the ε subunit of AP-4) KO mice display a thin corpus callosum and axonal swellings in various brain regions and spinal cord, such as hippocampal and Purkinje neurons, with a defect in autophagosome maturation. In addition, AP-4 ε KO mice also display impaired performance on the rotarod, and other motor and behavioral abnormalities, such as reduced grip strength, hindlimb clasping, increased ambulation, and enhanced acoustic startle response [210,211]. The disruption of the gene encoding the β subunit of AP-4 can increase the accumulation of axonal autophagosomes that contain AMPAR and transmembrane AMPA receptor regulatory proteins (TARPs) in axons of cerebellar Purkinje cells and hippocampal neurons both in vivo and in vitro. In fact, AP-4 is associated with TARPs that tightly bind to all subunits of AMPA receptors GluR1-GluR4, whereby the mutation of AP-4 may regulate the proper somatodendritic-specific distribution of AMPA receptor, TARP, and other cargo in different neurons (such as the hippocampus, spinal cord, and other brain regions) [204,211]. Moreover, knockouts of Atg5 and Atg7 in Purkinje neurons and rhodopsin neurons of the retina, and Atg7 in agouti-related peptide (AgRP) neurons of the hypothalamus have also been well-studied [212,213,214,215,216]. For instance, the disruption of the genes Atg5 and Atg7 caused Purkinje axonal swellings, followed by progressive dystrophy and degeneration in the axon [212,213]. Thus, the above studies suggest that autophagy deficiency induced by impairing positive regulators Ap4 (or ATG5, or ATG7) contributes to neuroaxonal dystrophy, axonal degeneration, and abnormal neuron functions, such as AMPAR-mediated synapse efficiency, particularly in Purkinje and hippocampal neurons.

However, Ban et al. reported that the inhibition of autophagy by atg7 small interfering RNA (siRNA) resulted in the elongation of axons, while activation of autophagy by rapamycin suppressed axon growth in cortical neurons [217]. Consistent with this, the mutation of Mir505-3p, which is an autophagic regulator for axonal elongation and branching, consistently enhanced the autophagy signaling cascade for axon growth and autophagosome biogenesis [218]. So, why does knocking out a positive regulator of autophagy has the opposite effect? We note that the mammalian central nervous system needs autophagy to maintain its normal functions and homeostasis, particularly in the axons [163,165,168,219]. Different brain regions have distinct types of neurons, synapse connections, and biological functions, which depend on dissimilar autophagy needs in different neurons [165,168]. The deletion of positive regulators (such as AP-4, ATG5, and ATG7) displays axonal degeneration in different neurons of brain regions, particularly in hippocampal neurons, whereas the deletion of positive regulators (such as ATG7) in cortical neurons displays axon growth. Thus, this apparent paradox may suggest that different types of neurons (such as hippocampal, Purkinje, hypothalamic, and rhodopsin neurons) have different structures for specific neuron functions, and different neuron types have dissimilar autophagy needs during axon regulation, which results in different effects for deletion of positive regulators.

In contrast, Bassoon and Piccolo negatively regulate autophagy by inhibiting autophagosome biogenesis. Specifically, Bassoon is a scaffold protein in the presynaptic active zone and interacts, sequesters, and prevents ATG5, an E3-like ligase that is essential for the binding of LC3 to PAS membranes, actively interfering in the formation of autophagosomes [220]. Functional loss of Bassoon in primary embryonic hippocampal neurons leads to the accumulation of LC3-positive complexes and increased autophagy activities at axon terminals, and impaired synaptic integrity. Yet, it is accompanied by reduced numbers of synaptic vesicles, implicating that presynaptic autophagy still actively hydrolyses synaptic vesicles as well as their components in the absence of Bassoon [221]. Moreover, the ubiquitin ligase RPM-1 restricts UNC-51/ULK kinase activity, and inhibits the formation of autophagosomes for axon termination, negatively regulating the initiation of autophagy, synapse maintenance, and behavioral habituation (Figure 8a) [222].

Additionally, the brain-derived neurotrophic factor (BDNF) is not only an important mediator of activity-dependent modifications of synaptic strength in synapses but also a key regulator of autophagy (Figure 8a). BDNF increases the accumulation of synaptic vesicles at active zones and the quantal neurotransmitter release within the presynaptic terminal, and specifically suppresses axonal autophagy by inhibiting autophagosome biogenesis [223]. The conditional deletion of BDNF exhibits an overabundance of autophagosomes in the axon terminals, as confirmed by electron microscopy in the hippocampal neurons [224]. Taken together, these shreds of evidence indicate that autophagy directly regulates axon growth and presynaptic neurotransmitter release, via the mechanisms that remain to be clearly elucidated.

### 5.2. Autophagy in Dendrites

Dendrites are the foremost sites for information input and integration from presynaptic neurotransmitters. Function impairments of dendrites lead to abnormal neuronal activities and neuronal circuits. Under native conditions, autophagy functions as a scavenger to clear cytotoxic materials and recycle the degraded components, especially in dendrite spine pruning [225]. Autophagic dysfunction has been linked to dendritic atrophy as well as neuronal cell death. However, little is known about the underlying mechanisms of dendrite autophagy in neurons.

In Drosophila, basal ATG genes direct dendritic terminal branching and arborization by positive regulation of the homeodomain transcription factor Cut. The genetic disruption of Atg genes boosts type-specific neuronal dendritic arborization, whereas the function gain of these genes dramatically reduces the dendritic complexity in multidendritic sensory neurons, such as deficient dendritic branching and decreased arbor growth [226]. In addition, the specific deletion of ATG7 in dopaminergic neurons results in delayed dendritic neurodegeneration, accompanied by large dendritic ubiquitinated inclusions and dendritic dystrophy [227]. Whole-brain knock-out of Atg7 significantly impairs dendritic spine pruning, promoting social behavioral disorders in mice [228]. These results show that constitutive autophagy is essential for dendrite arborization, branching, and spine fine-pruning.

Moreover, dendritic autophagy also mediates the intracellular trafficking of postsynaptic membrane receptors. Previous studies have suggested that the lapidated GABARAP subfamily of mATG8s facilitates intracellular trafficking of Gamma-aminobutyric acid A (GABAA) receptor, and increases the GABAA receptor surface expression in primary hippocampal neurons, PC12 cells, and Xenopus oocytes (Figure 8b) [183,229]. Dendritic autophagy degrades the GABAA receptor at postsynaptic terminals in the absence of presynaptic innervation, with the diffuse distribution of the GABAA receptor on the plasma membrane of the muscle. Genetic disruption of autophagy rescues neurotransmission defects in non-innervated muscles. Interestingly, in *C. elegans*, dendritic autophagy reduces the surface expression of the GABAA receptor in non-innervated muscles [230]. The deletion of autophagy does not increase acetylcholine (ACh) currents, accompanied by non-colocalization with autophagosome markers [231]. These results indicate that autophagy selectively may modulate neuronal excitation and inhibition.

Autophagy also coordinates synaptic activity in the dendrites [191]. In cultured hippocampal neurons, KCl depolarization and brief low-dose N-methyl-D-aspartate (NMDA) transiently increases the number of LC3-II puncta and autophagosomes in spines and accelerates the degradation of α-amino-3-hydroxy-5-methyl-4-isoxazolepropionic acid (AMPA) receptors, which are partially recovered by the NMDA receptor inhibitor APV. The short hairpin RNA-induced knockdown of ATG7 and the autophagy inhibitor wortmannin inhibits chemical-LTD-mediated autophagy through the PI3K-Akt-mTOR pathway [232]. Additionally, BDNF functions as a chemical inducer of LTP and modulates dendritic autophagy in hippocampal neurons. BDNF deficiency leads to LTP deficits and memory impairment in mice, which are alleviated by the suppression of autophagy [224]. Several scaffolding proteins including SHANK3, PSD-95, and PICK1 in dendrites are also colocalized in autophagosomes, and their degradation may regulate synaptic plasticity (Figure 8) [233].

### 5.3. Autophagy in Soma

Little is known about neuronal autophagy in the soma. Previous evidence indicates that autophagosome biogenesis events are enriched in the distal axon, and are infrequently observed in the soma of the hippocampal cultures and the primary dorsal root ganglion via live-cell imaging [183]. The soma in the hippocampal neurons contains the bulk of autophagosomes, which are defined by differences in dynamics as well as maturation state and are generated from different neuronal compartments and are enriched by populations of proteolytically active lysosomes enabling cargo degradation [234]. The autophagosomes derived from the neuronal soma tend to cluster and are less mobile.

The most common form of autophagy in neuronal soma is mitophagy. Mitochondria play a key role in neuronal function and survival, and mitophagy is essential for eliminating damaged and aged mitochondria. The Parkin-mediated and mitochondrial Δψm-induced mitophagy in cortical neurons is the unique process where the Parkin-labeled mitochondria are accumulated at a somatodendritic site and subsequently degraded by the autophagy-lysosomal pathway in soma [235]. In addition, the functional deletion of genes in the PINK1/Parkin pathway results in diminished mitochondrial membrane potential and compromised somatic mitochondrial integrity, but does not increase the mitochondrial density or length in axons, suggesting that mitophagy-dependent mitochondrial turnover events may be restricted to the neuronal soma in vivo [236]. Overall, several studies have shown that autophagy in soma directly regulates mitochondria quality control and involves neuronal senescence and multiple neurological disorders, by mechanisms that remain to be further elucidated.

### 5.4. Neuronal Autophagy and Synaptic Plasticities

Synaptic plasticity, the activity-dependent change in the efficacy of synaptic connection, has long been thought to represent a fundamental property of learning and memory and an important component of brain adaptation [237]. Synaptic plasticity processes rely heavily on the quality control of protein-protein interactions and organelles, and autophagy is crucial to the homeostatic maintenance of synaptic plasticity, spine pruning, and axon guidance [238]. Recent studies indicated that autophagy plays a key role in information encoding, memory processing, and cognitive functions. When dysregulated, autophagy is associated with disrupted synaptic plasticity, neurological disorders, such as autism spectrum disorders and stroke, and neurodegenerative diseases, such as Alzheimer’s disease, Parkinson’s disease, multiple sclerosis, and others [239]. However, little is known about how autophagy regulates synaptic plasticity.

Loss of Atg7 in mouse neurons and microglial cells damages synaptosome degradation, results in larger axonal profiles, increases dendritic spines, and alters the strength of synaptic connection, similar to the results of cultured Atg7-deficient hippocampal neurons [228,240]. In animal models, autophagy-dependent changes in synaptic plasticity elicit a series of behavioral phenotypes, including learning and memory deficits, anxiety-like behaviors, social-behavioral defects, and cognitive deficits [240,241,242,243]. For instance, conditional knockdown of autophagy-related genes including BCN1, Atg12, and FIP200, or pharmacological inhibition of autophagy-related activities in mouse hippocampal neurons impairs the behavioral performance in contextual fear conditioning and novel object recognition tasks, implying the role of autophagy in learning and memory [241].

There is growing evidence for autophagy in LTP and LTD, two major forms of synaptic plasticity. Autophagy activator rapamycin can block late LTP induced by theta burst stimulation in the mouse cortex, leading to impaired synaptic plasticity [244]. In addition, some evidence suggests that the pharmacological inhibition of neuronal autophagy with Spautin-1 also administers the induction of LTP in CA1, whereas autophagy inhibition with BDNF is found to permit LTP in hippocampal regions [224,241]. These contradictory results may be caused by different animal backgrounds, surgical procedures, animal age, side effects of molecular drugs, or other unknown mechanisms.

Hyperactivation of mTORC1 by delivery of shRNA into the CA1 results in decreased autophagy and increased dendritic spine density with an aberrant morphology, and exaggerated LTD in mouse models of fragile X syndrome [243]. The induction of NMDAR-dependent LTD triggers a modified reorganization of the postsynaptic scaffolding protein PSD-95. This process is driven by the autophagic machinery that removes the T19 phosphorylated form of PSD-95 in the synaptic cleft, increasing the surface mobility of the glutamate receptor AMPAR, thereby attenuating synaptic efficacy and increasing short-term plasticity [233]. Notably, several key kinases regulating autophagy activity are also associated with synaptic plasticity, such as AMPK, Akt, and mTOR [245]. In brief, autophagy alters the abundance of selective synaptic proteins in a spatiotemporal-dependent manner, ultimately enabling fine-tuning of synaptic structure, function, and plasticity.

## 6. Epilepsy and Autophagy

### 6.1. Epilepsy

Epilepsy is a serious chronic neurological disease characterized clinically by recurrent, unprovoked epileptic seizures, which lead to cognitive decline, temporary neurological dysfunction, brain damage, accidental injury, and other consequences [246]. A survey of international studies showed that the prevalence globally is 6.38 per 1000 persons for active epilepsy, and 7.60 per 1000 persons for lifetime prevalence [247]. The annual incidence rate of epilepsy is 61.44 per 100,000 persons, while the long-term recurrence is 83.6% at 10 years [247,248]. Similarly, epilepsy is associated with patient stigma, neurological and psychiatric comorbidities, and high medical costs, seriously affecting the patient’s health and life quality, particularly in low-income countries.

Seizures are considered as a transient occurrence of clinical symptoms or signs due to synchronous, high-frequency, or abnormal excessive neuronal firing activity in the central nervous system, and actually are caused by multiple syndromes of different etiologies, which are related to congenital or acquired brain malformations, structural injuries or lesions, cerebral cysticercosis, and other brain diseases or disorders [249,250]. During epileptogenesis, the patient’s brain manifests an enduring and pathologic tendency to have spontaneous, recurrent seizures, facilitating neuronal hyperexcitability and abnormal synaptic plasticity [251]. The process is capable of disrupting normal neuronal information processing, the imbalance between excitatory and inhibitory neuronal networks and other networks. For example, epileptogenic networks in generalized epilepsies involve bilateral thalamocortical structures and are widely distributed in the brain, while focal epilepsies implicate the neural circuits in one cerebral hemisphere, commonly the neocortical or limbic cortex [249]. Despite breakthroughs having been made in the genetic basis of most generalized epilepsies and structural cerebral abnormalities in focal epilepsies, the underlying pathophysiological mechanisms remain incompletely understood [252,253].

### 6.2. Neuronal Autophagy and Epilepsy

The mesial temporal sclerosis in epilepsy patients is the best-ascertained lesion, which is characterized by neuron loss in specific hippocampal subfields, collateral axonal sprouting, synaptic reorganization of neural circuits, and functional and structural alterations in the glial [254]. Autophagy is thought to be a potential mechanism for these processes or targets for disease-modifying therapies. Most studies suggest that neuronal mTOR hyperactivation, which is mediated by loss-of-function gene mutations of mTOR inhibitor proteins including phosphatase and tensin homolog (PTEN), TSC1, TSC2, and STE20-related kinase alpha (STRADα), links to the severity of epilepsy in an animal model of epileptogenic cortical malformations, TSC, and systemic lupus erythematosus, the most common genetic etiology of epilepsy and neurobehavioral disabilities [255,256,257]. Some evidence also exhibits that the mTOR signal participates in temporal lobe epilepsy (TLE), chemoconvulsive compounds-mediated experimental epilepsy, and other epileptogenesis forms of genetic or acquired epilepsy, such as traumatic brain injury and Lafora disease (LD) [258] (Table 1). Inhibition of mTOR activity is capable of preventing epilepsy-induced neuronal alterations and the development of epilepsy [256]. Recent evidence supports the solid correction between mTOR-dependent autophagy and multiple seizure models and epilepsies [258]. However, although the close relationship between mTOR signal and autophagy activity has been automatically validated by many studies, mTOR plays a variety of roles in neurogenesis, neuronal development, synaptic plasticity, neuronal excitability, protein expression of different neuronal molecules, and other metabolic activities [259]. These may induce atypical neuronal circuits eliciting epileptogenesis independently from autophagy impairments. Thus, the overemphasized effects of mTOR for autophagy activity may mislead the interpretation of mTOR-related experimental results, and ignore the autophagy-unrelated contribution of mTOR to epilepsy.

In fact, reliable conclusions may only be drawn from studies that directly assess the autophagy condition associated with epileptogenesis and epilepsy-induced brain alterations. A recent study shows that autophagy markers p62 and LC3, which usually represent the status of autophagy activation, exhibit an obvious increase of autophagy flux in autosomal dominant lateral temporal epilepsy, which can be incompletely rehabilitated by small-molecule correctors of autophagy-related proteins Reelin [260]. Nevertheless, autophagy markers are also found to increase on the blockade of autophagic flux and impairment of autophagy progression. The paradoxical increase of autophagy markers in the context of autophagy disruption is the case for mTOR-independent autophagy inhibition, which should also be taken into account when monitoring the autophagic status of epilepsy [261]. In static encephalopathy of childhood with neurodegeneration in adulthood (SENDA), *de novo* mutations of the autophagy-related gene *WIPI4* (also known as *WDR45*) severely impair its protein expression, and they lower the autophagic activity with the aberrant accumulation of LC3-positive structures at an early stage of autophagosome formation, which directly contributes to the occurrence of encephalopathic seizures [262]. In addition, the conditional ablation of Atg7 in mouse forebrain neurons induces the development of spontaneous seizures [263]. These two studies demonstrate that the blockage of the autophagy flux and the accumulation of aberrant autophagic structures occur along with lower autophagy activity in epilepsy, but with normal mTOR signaling.

Consistently, research on epilepsy animal models also shows the increased ratio of LC3 II to LC3 I and decreased p62 protein levels after epileptic seizure onset, reflecting an abnormal increase in autophagic activity in kainic acid (KA)-induced rats and mice [264,265]. This may trigger chronic hyperactivation of glutamate receptors, slow degradation of GABAA receptors, and formation of protein aggregates, and in turn, further exacerbate neuronal damage in epilepsy animals, creating a vicious cycle between autophagy and epilepsy [266,267].

Some genetic sequencing studies for epilepsy patients and animal models suggest that the mutations in *TBCK* (encoding the TBC1-domain-containing kinase and inhibiting mTORC1 signaling), *VSP15* (perturbing endosomal-lysosomal trafficking and autophagy flux), *EPG5* (encoding a tethering factor that regulates the specific fusion of APs with late lysosomes/endosomes), *SNX14* (encoding the sorting nexin family protein that mediating lysosome-AP fusion), *ATP6V1A* (encoding for the “A” subunit of the v1 sub-complex of V-ATPase, affecting lysosomal homeostasis and autophagy), *ATP6AP2* (encoding a key regulator of v-ATPase, and its loss leads to lysosomal and autophagic defects), *DMXL2* (encoding a member of the WD40 protein family that regulates v-ATPase trafficking and activity), or *FAM134B* (encoding the ER autophagy receptor) may contribute to the occurrence of epilepsy in space- and time-dependent manners [268,269] (Table 1). Notably, recent studies have demonstrated that microRNAs (miRNAs) and long noncoding RNAs (lncRNAs) involve the post-transcriptional regulation of autophagy-related proteins and the development of epilepsy pathophysiologies, such as miR-101, miR-181b, miR-134, miR-142, miR-421, miR-223, and Zinc finger antisense 1 (ZFAS1), metastasis-associated lung adenocarcinoma transcript 1 (MALAT1), respectively [270,271,272,273,274,275]. Moreover, glia including astrocytes and microglia also plays a multi-faceted role in autophagy-mediated mechanisms that determine seizures and epileptogenesis [276,277].

Collectively, current research suggests that abnormal autophagy occurs in epilepsy, and alters the physiological state of certain neurons [258,263,265,266,278,279,280,281] (Table 1). These state anomalies may encompass the changes at the molecular and cellular levels as well as at the circuit level [11,258,263,265,279,281,282]. For instance, Changes in cell signaling (such as mTOR and Atg7) alter the neuron’s overall ability to receive, integrate, and process inputs from different types of molecular cues, and the synapse morphology and function that decrease or increase the neuron’s ability to stimulate its synaptic partners [11,258,263,268,279,281,282]. Aberrant autophagy in epilepsy also alters the expression or function of ion channels (such as the GABAA receptor or glutamate receptor), resulting in decreased or increased neuronal excitability, and therefore decreased or increased ability to conduct action potentials or other electrical signals [11]. Rapid mobilization of neuronal growth processes that establish new synaptic connections or neurite retraction that removes existing connections may be disrupted by abnormal autophagy [279,280,282]. However, the underlying molecular mechanisms by which autophagic activity regulates these alterations to promote epileptogenesis remain unclear.

**Table 1 cells-11-02621-t001:** The autophagy-related molecular changes in epilepsy animal models and patients.

Models/Patients	Molecular Changes	Pathological Changes	References
Lafora disease mice	Increased Rab5, p62 protein level, decreased LC3-II levels	Generalized stimulus-sensitive tonic-clonic seizures	Puri and Suzuki., 2012 [283]; Criado et al., 2012 [284]
Pilocarpine-induced model mice	Increased levels of Beclin 1, ATG5, ATG7 and the ratio of LC3II/I	Epilepsy	Ying et al., 2020 [285]
N-ethyl-N-nitrosourea (ENU)-induced mice mutants	*Vps15* mutation, decreased LC3-II/LC3-I ratio	Cortical atrophy, dysplasia, and epilepsy	Gstrein et al., 2018 [286]
TSC1/PTEN KO mice	mTOR hyperactivation, increased Ulk1 phosphorylation	Epileptogenesis	Yasin et al., 2013 [287]
Kainic acid treatment mice	Increased LC3-II levels, elevated ratios of phospho-mTOR/mTOR	Repeated seizures	Shacka et al., 2007 [265]
Atg7 KO mice	p62 accumulation	Spontaneous seizures	McMahon et al., 2012 [263]
Depdc5 KO mice	Increased mTORC1 signaling	Spontaneous seizures	Yuskaitis et al., 2018 [288]
PTEN KO mice + mTOR inhibition	Decreased mTOR activity	Decreased the seizure frequency and death rate	Kwon et al., 2003 [289]
Pilocarpine-induced model rats	Increased LC3-II/LC3-I ratio and beclin1 level	Status epilepticus	Cao et al., 2009 [290]
Kainic acid treatment rats	mTOR activation	Status epilepticus	Macias et al., 2013 [291]
Kainic acid treatment rats + rapamycin	Decreased mTOR activity	Reduced epilepsy	Zeng et al., 2009 [292]
Pilocarpine-induced model rats	mTOR activation	Status epilepticus	Buckmaster et al., 2009 [279]
Pilocarpine-induced model rats + rapamycin	Decreased mTOR activity	Reduced seizure activity	Huang et al., 2010 [293]
Infantile spams/West syndrome rats	mTORC1 pathway overactivation	Spasms, epileptic encephalopathies	Raffo et al., 2011 [294]
*VPS15* mutation in humans	p62 accumulation	Cortical atrophy, late-onset epilepsy	Gstrein et al., 2018 [286]
Beta-propeller protein-associated neurodegeneration patients	*De novo* mutation in *WDR45*	Developmental and epileptic encephalopathies	Carvill et al., 2018 [295]
Autosomal dominant lateral temporal epilepsy patients	*Reelin* mutation	Epilepsy	Dazzo and Nobile., 2022 [260]
Vici syndrome patients	*EPG5* mutation	Severe seizure disorder, progressive neurodegeneration	Byrne et al., 2016 [296]
Pediatric-onset ataxias patients	*SNX14* mutation	Progressive cerebellar neurodegeneration, developmental delay, intellectual disability, and seizures	Akizu et al., 2015 [297]
Ohtahara syndrome patients	*DMXL2* mutation	Intractable seizures and profound developmental disability	Esposito et al., 2019 [298]
Children with *TBCK p.R126X* mutations	Increased LC3-II/LC3-I ratio	Focal and generalized seizures	Ortiz-González et al., 2018 [299]
Epilepsy patients	*ATG5* gene variant, *ATP6V1A/ ATP6AP2* mutation, increased Beclin1 expression	Late-onset epilepsy, temporal lobe epilepsy	Zhang et al., 2021 [300]; Van Damme et al., 2020 [301]; Hirose et al., 2019 [302]; Yang et al., 2022 [303]
Focal cortical dysplasia in childhood	mTOR activation, p62 accumulation, *TSC1/TSC2* mutation	Epilepsy	Yasin et al., 2013 [287]
Human TSC patients	Increased in Ulk1 phosphorylation, p62 accumulation	Cognitive dysfunction, early-onset, intractable epilepsy	McMahon et al., 2012 [263]
Hippocampal neuronal culture model of acquired epilepsy	Elevated LC3-II/LC3-I ratio	Acquired epilepsy	Xie et al., 2020 [269]

## 7. Conclusions

Converging lines of research support a model whereby all cells, including invertebrate animals, insects, yeasts, and cultured cells, have evolved highly specific mechanisms to ensure protein quality control and maintenance of cell homeostasis, namely autophagy. These mechanisms involve the biogenesis and trafficking of the autophagosome, formation, and degradation of the autolysosome, and intricate regulation of autophagy-related molecular proteins in space- and time-dependent manners, particularly in the neuron. Recent findings have driven forward our understanding of how neuronal autophagy regulates the growth and development of axons and dendrites and influences spine pruning, neurotransmitter release, synapse plasticity, and other neuronal functions. Structural and functional integrity in neurons is more sensitive to changes in autophagy than that of non-neuronal cells due to the highly specialized compartments necessary for intercellular communications. Autophagy impairments likely induce accumulation of toxic protein aggregates and worn organelles, as well as interruption of synapse pruning, ultimately affecting the normal physiological function of the neuron. In the past decades, autophagy alterations have begun to be applied to the field of epileptogenesis besides neurodegenerative disorders. Recent studies have suggested that hyperactivation of the mTOR signaling pathway and abnormal autophagy activity occur in different epilepsy animal models and epilepsy-related patients, whereas inhibition of mTOR activity can ameliorate seizures in various epilepsy models. For instance, the special deletion of *TSC1* or *PTEN* genes in mouse neurons results in seizures and compromised autophagy activity, consistent with human TSC patients. Genetic sequencing of multiple types of epilepsy patients and seizure models indicated that gene mutations in autophagy-related pathways may contribute to the occurrence of epilepsy, such as *TBCK*, *EPG5*, *VSP15*, etc. In addition, impaired autophagy is sufficient to induce epilepsy in rodent models, such as genetic inactivation of Atg7 in mice. However, it represents an underexplored research avenue for epilepsy. There are still many important concepts, principles, and issues to be addressed in this field. Promising directions involve working out the details, including seeking: (1) to decipher how synapses signal membrane biogenesis and autophagy initiation, (2) to uncover how neuronal autophagy and synaptic efficacy are cross-regulated in seizures, (3) to uncover the interconnection of autophagy activity with antiepileptic drugs, and (4) to define the diversity and dynamic nature of the autophagic cargo of different neuronal compartments in terms of their physiological states and epilepsy pathologies.

## Figures and Tables

**Figure 1 cells-11-02621-f001:**
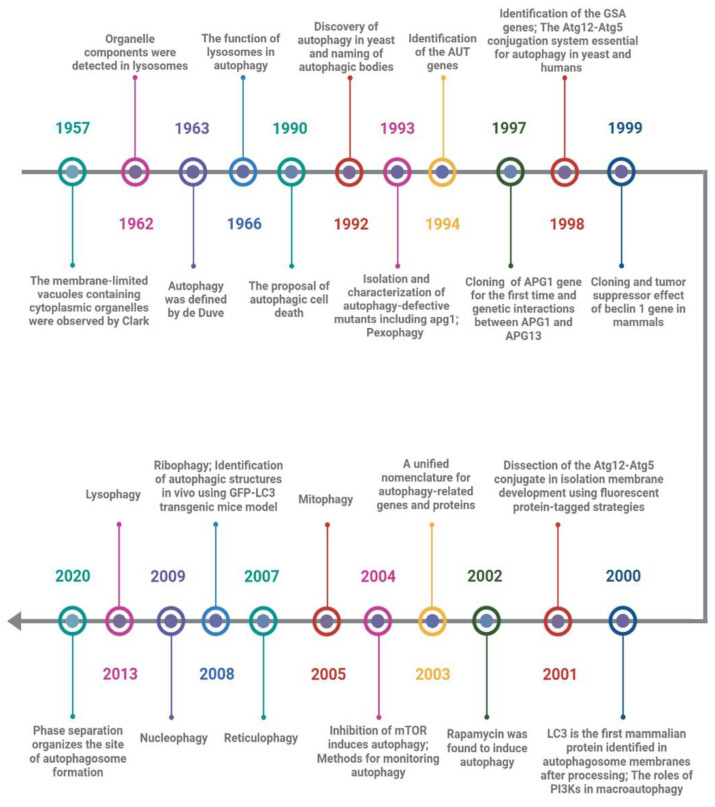
Timeline of historical events in autophagy. This timeline depicts the key discoveries of autophagy. Unfortunately, it is not possible to include all discoveries due to limited space.

**Figure 2 cells-11-02621-f002:**
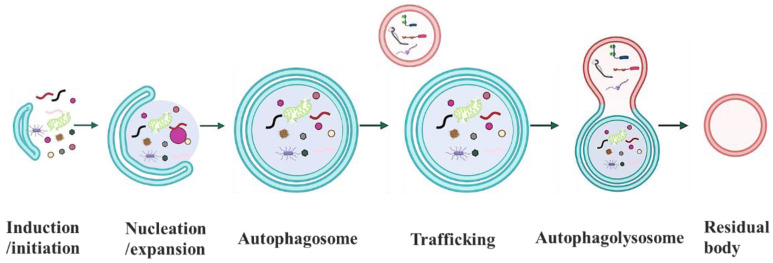
Schematic for macroautophagy. Macroautophagy is a multistep process that involves the induction/initiation, nucleation/expansion, maturation, trafficking of autophagosome, formation of autophagolysosome, and release and reuse of degradation products.

**Figure 3 cells-11-02621-f003:**
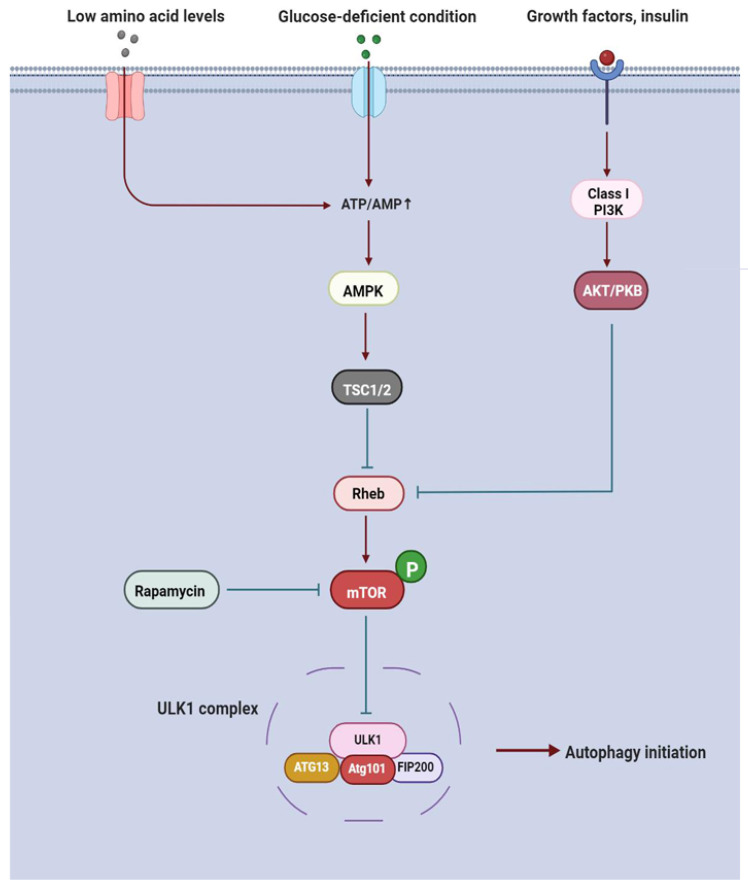
Overview of upstream signaling pathways that regulate macroautophagy. Autophagy induction is initiated when mTOR signaling is suppressed by a range of stress modalities and an upstream signaling cascade. Abbreviations: PKB, protein kinase B; AMPK, AMP-activated protein kinase; TSC1, tuberous sclerosis complex 2; TSC2, tuberous sclerosis complex 2; mTOR, mammalian target of rapamycin.

**Figure 4 cells-11-02621-f004:**
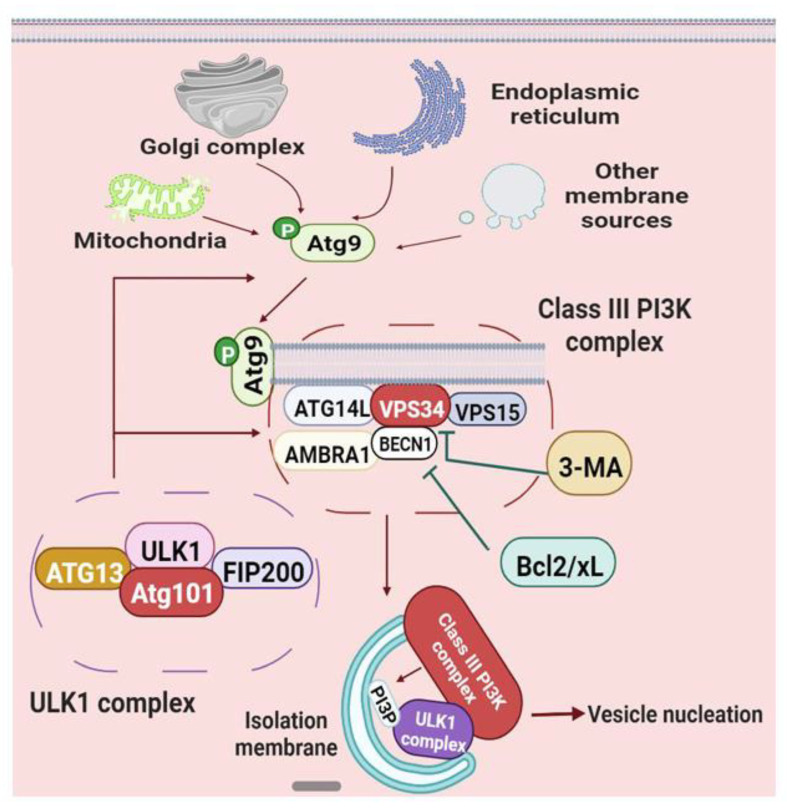
Schematic diagram of vesicle nucleation in macroautophagy. Vesicle expansion is orchestrated by Atg proteins and complexes, such as the class III PI3K complex. Abbreviations: 3-MA, 3-methyl adenine.

**Figure 5 cells-11-02621-f005:**
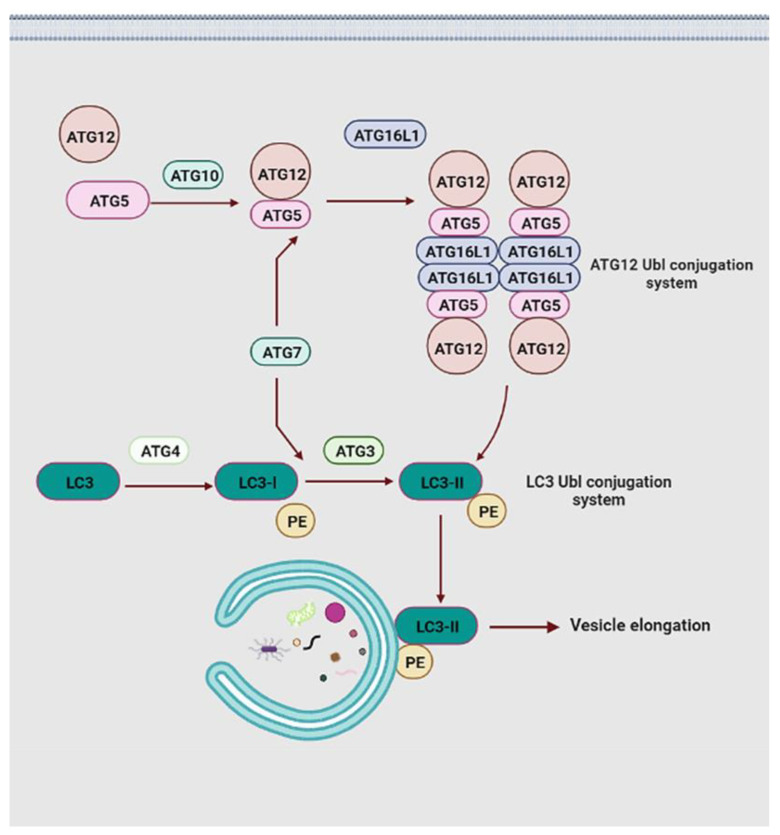
Schematic representations of vesicle elongation events. The elongation of the isolation membrane is mainly directed by two ubiquitin-like protein conjugation pathways. Abbreviations: PE, phosphatidylethanolamine.

**Figure 6 cells-11-02621-f006:**
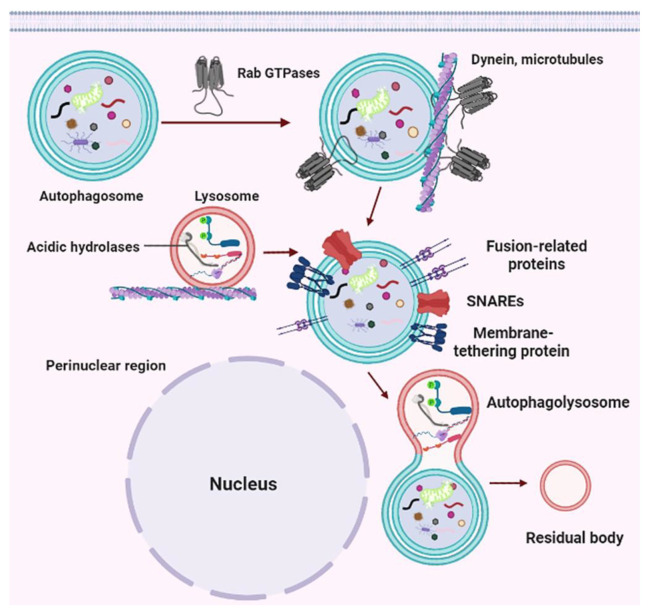
Schematic for the trafficking of autophagosome, formation of autophagolysosome, and release and reuse of degradation products in macroautophagy. Abbreviations: SNAREs, the soluble N-ethylmaleimide sensitive fusion protein attachment protein receptor.

**Figure 7 cells-11-02621-f007:**
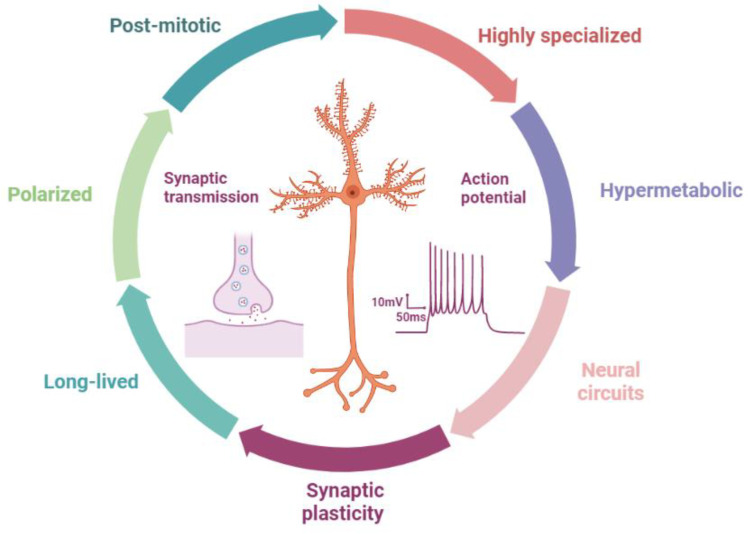
Neurons have unique architectures and characteristics.

**Figure 8 cells-11-02621-f008:**
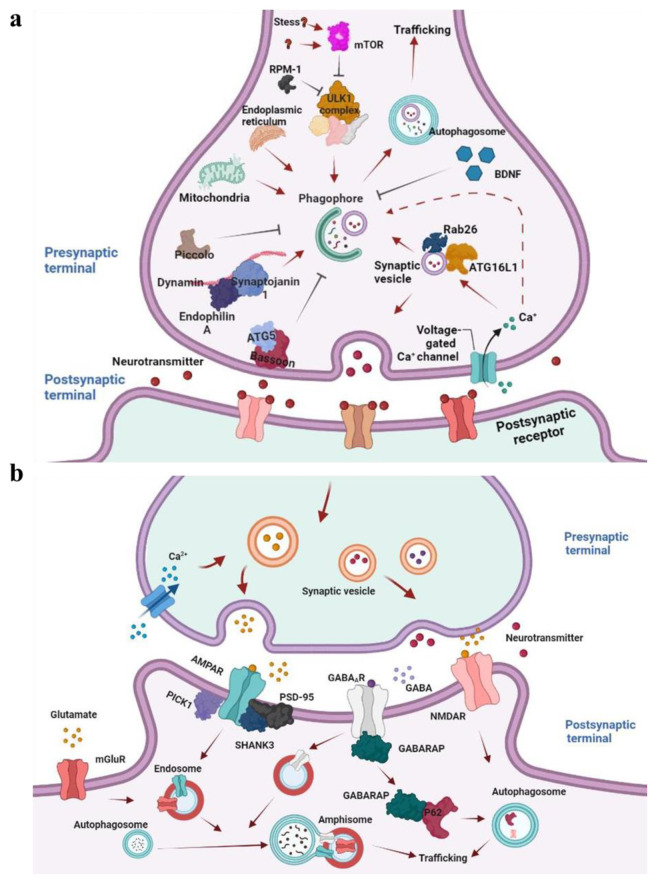
The cross-regulation between neuronal autophagy and synaptic vesicles at the pre-synaptic and post-synaptic terminals. Neuronal activity induces autophagosome formation at the presynaptic terminal by regulating several presynaptically enriched adaptors (such as Synaptojanin1, Endophilin A, and Bassoon) and regulatory proteins (**a**). At the postsynaptic terminal, neuronal activity also upregulates neuronal autophagy, resulting in endocytic removal of neurotransmitter receptors (such as GABAAR, NMDAR, and AMPAR) from the plasma membrane, presumably by modulating receptor adaptor proteins (**b**).

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
