# Peer review of "Molecular Mechanism and Regulation of Autophagy and Its Potential Role in Epilepsy"

_cells, 2022, doi:10.3390/cells11172621_

Round 1
Reviewer 1 Report
The authors of this manuscript described the molecular mechanism of autophagy and the role of autophagy in neurons and epilepsy. Generally, the authors described well and in detail. And this review manuscript is well organized. Especially, the authors summarized recent findings on autophagy in neurons and epilepsy. There are small suggestions to improve this manuscript.
1. It would be great to be added one more figure to show "the historical landmarks of autophagy". For instance, using an arrow and put historical landmarks with specific years and contents.
2. In line 55, the reference should be added.
3. For "the historical landmarks of autophagy", I carefully suggest two more landmarks. 1) the findings of organelle-specific autophagy and 2) autophagic cell death.
4. In line 138, the study regarding the resources for autophagosomes is intensively studied. There are several candidates for sources of autophagosomes such as ER, Golgi, plasma membranes, and mitochondria.
5. In line 32 and several parts in the manuscript, chaperone-mediated autophagy is abbreviated as CMV. But it should be corrected to CMA. And in line 237, the central coil domain stands for CCD. But CCD is the abbreviation of "coiled-coil domain".
6. Some of the typos should be corrected. Such as line 191, "TSC1and" to "TSC1 and", line 280, "LC3- I" to "LC3-I".
7. In line 258, it would be good to add one more reference "PMID 21062745" Sun et al., 2011 JBC. It is the original paper to show the inhibition effect of Rubicon in VPS34 activity.
Author Response
Comments and Suggestions for Authors: The authors of this manuscript described the molecular mechanism of autophagy and the role of autophagy in neurons and epilepsy. Generally, the authors described well and in
detail. And this review manuscript is well organized. Especially, the authors summarized recent findings on autophagy in neurons and epilepsy. There are small suggestions to improve this manuscript.
Point 1: It would be great to be added one more figure to show "the historical landmarks of autophagy". For instance, using an arrow and put historical landmarks with specific years and contents.
Response 1: Thank you for your valuable comments. This idea is great. We have added a new figure (in lines 110-113) to the revised manuscript.
Point 2: In line 55, the reference should be added.
Response 2: Thanks for your kind suggestion. The reference has been edited according to your suggestion (in line 55).
Point 3: For "the historical landmarks of autophagy", I carefully suggest two more landmarks. 1) the findings of organelle-specific autophagy and 2) autophagic cell death.
Response 3: Thank you for your valuable suggestions. This advice is very valuable. We have added these to the latest manuscript (in lines 75-77 and 128-131).
Point 4: In line 138, the study regarding the resources for autophagosomes is intensively studied. There are several candidates for sources of autophagosomes such as ER, Golgi, plasma membranes, and mitochondria.
Response 4: Thank you for your kind advice. We have corrected this incorrect description regarding the resources for autophagosomes (in lines 145-147) in the latest manuscript.
Point 5: In line 32 and several parts in the manuscript, chaperone-mediated autophagy is abbreviated as CMV. But it should be corrected to CMA. And in line 237, the central coil domain stands for CCD. But CCD is the abbreviation of "coiled-coil domain".
Response 5: Thank you for your professional advice. We have corrected both descriptions in the latest revision (in lines 32 and 254-255).
Point 6: Some of the typos should be corrected. Such as line 191, "TSC1and" to "TSC1 and", line 280, "LC3- I" to "LC3-I".
Response 6: Thank you for your kind suggestion. We have addressed this point in the revised manuscript (in lines 204-205 and 296-297).
Point 7: In line 258, it would be good to add one more reference "PMID 21062745" Sun et al., 2011 JBC. It is the original paper to show the inhibition effect of Rubicon in VPS34 activity.
Response 7: Thank you for your valuable advice. We have added this reference in the latest manuscript (in line 275).
Dear Reviewers,
On behalf of all the contributing authors, I would like to express our sincere appreciation for your constructive comments concerning our article entitled “Molecular mechanism and regulation of autophagy and its potential role in epilepsy” (Manuscript ID: cells-1850560).
These comments are all valuable and helpful for improving our article. According to your professional and valuable comments, we have made extensive modifications to our manuscript to make our results convincing. In this revised version, changes to our manuscript were all highlighted within the document by using the “Track Changes” function of MS Word. Point-by-point responses to two reviewers are listed below this letter (Please see the attachment).
Thank you again for your positive comments and valuable suggestions to improve the quality of our manuscript. We have tried our best to improve and made some changes to the manuscript.
Best regards
Sincerely yours
Han-xiao Zhu
The First Affiliated Hospital of Dali University,
32 Carlsberg Avenue, Dali City 671000, Yunnan, China,
e-mail: zhu_hanxiao1109@163.com

Reviewer 2 Report
The manuscript compiles the most recent research on neuronal autophagy and try to connect autophagy impairment with epilepsy. The manuscript is well written and organized. The parts of autophagy and neuronal autophagy are well documented and provide a general overview of macroautophagy as the major type of autophagy. Then, the manuscript highlights the most important molecular players in neuronal autophagy in different compartments of a neuron: soma, dendrites and axons.
However, the major concerns in the manuscript rise in the part regarding the connection between neuronal autophagy and epilepsy. From the manuscript, it seems that there is no relevant evidence of impairment of neuronal autophagy in epileptic patients. Furthermore, the description of research on autophagy in animal models of epilepsy is scarce. By reading the section of autophagy in different neuronal compartment, the reader may expect to find evidence of impairment of autophagy in dendrites, axons and/or soma of neurons in animal models of epilepsy. However, there is no evidence of autophagy impairment in these compartments in any animal model of epilepsy or in samples from epileptic patients.
Here are some other comments that authors may want to address to further increase the quality of their manuscript:
Line 143: “CMV was the first discovered process by…” Do author meant CMA?
Font in figures and figures themselves are very small.
p62/SQSTM1 is an important player in the macroautophagy process as an intermediate cargo and a common autophagy marker; however, it is not well introduced in the manuscript.
The formation and expansion of a phagophore and the formation of an autophagosome are extensively described; however, the fusion of the autophagosome with lysosomes is barely mentioned. Authors may consider compensating the extension of the description of these steps of autophagy by shortening the text regarding the early steps and extend the text of the late autophagy steps.
Figure 2. There is a list of the architecture and characteristics of neurons. How autophagy regulates these characteristics? What of these neuronal characteristics are relevant to epilepsy?
Line 417: “This apparent paradox may suggest that different types of neurons have unique axon architectures, arrangements, orientations, and different neuron types have dissimilar autophagy needs during axon regulation.” This statement at the end of a paragraph is not well supported by the evidence described earlier. What type of neurons show autophagy impairment in axons?
Some statements do not have references, for example in lines 649-662
The first paragraph of the section 6.2 may fit better at the end of the section 6.1.
The term “autophagy flux” is not well introduced.
Line 624: “Consistently, research on epilepsy animal models also shows the increased ratio of LC3 II to LC3 I and decreased p62 protein levels after epileptic seizure onset, reflecting a blockage of autophagic flux…” This is not interpreted as a blockage of autophagy flux, but as a stimulation of autophagy flux. In addition, LC3-II/actin is a more accurate parameter than LC3-II/LC3-I ratio, as the detection of LC3-I is highly variable.
Including a table listing the studies on neuronal autophagy in animal models and in human samples will help better to understand the scientific progress in this field.
The Conclusion section does not clearly states what is known on neuronal autophagy and epilepsy.
Author Response
manuscript:
Point 1: Line 143: “CMV was the first discovered process by...” Do author meant CMA?
Response 1: Thank you for your valuable suggestion. Chaperone-mediated autophagy is abbreviated as CMV. We have corrected this abbreviation in the latest revision (in line 151).
Point 2: Font in figures and figures themselves are very small.
Response 2: Thank you for your valuable advice. We have redrawn the image and adjusted the font size in the revised manuscript (Figure 1 in lines 110-113; Figure 2 in lines 185-189; Figure 3 in lines 219-225; Figure 4 in lines 249-253; Figure 5 in lines 329-333; Figure 6 in lines 390-394; Figure 7 in lines 476-478; Figure 8 in lines 656-659).
Point 3: p62/SQSTM1 is an important player in the macroautophagy process as an intermediate cargo and a common autophagy marker; however, it is not well introduced in the manuscript.
Response 3: Thank you for your kind advice. We have added this in our latest manuscript (in lines 308-327).
Point 4: The formation and expansion of a phagophore and the formation of an autophagosome are extensively described; however, the fusion of the autophagosome with lysosomes is barely mentioned. Authors may consider compensating the extension of the description of these steps of
autophagy by shortening the text regarding the early steps and extend the text of the late autophagy steps.
Response 4: Thank you for your kind suggestion. We have added new descriptions regarding the late autophagy steps (in lines 339-364 and 369-412) in the revised manuscript.
Point 5: Figure 2. There is a list of the architecture and characteristics of neurons. How autophagy regulates these characteristics? What of these neuronal characteristics are relevant to epilepsy?
Response 5: Thank you for your professional comments. Unlike somatic cells, the neuron is a terminally differentiated cell (post-mitotic, polarized) and has highly polarized structures, including axons and dendrites, which are critical to their highly specialized functions (such as signaling transmission and synapse plasticity). Although the neuronal architectures and characteristics (such as hypermetabolic, long-lived, and highly specialized) rely on cell determination, autophagy is required for the maintenance of these properties and functions.
Firstly, the neuron is a long-lived and post-mitotic cell, whereby it has no capability to dilute the accumulated effects of damaged proteins, organelles, and themselves through mitosis. Autophagy has the capability to dominate the turnover of RNA, lipids, proteins, and organelles, which maintains neuronal survival throughout the lifetime of most organisms (Lee., 2012). Secondly, the neuron is a polarized cell and has highly polarized structures, such as axons and dendrites. Autophagy plays a key role in these aspects (such as complex
neural circuits and topographic maps).
For instance, during early growth and active developmental refinement of axons and dendrites (such as orphan neurites are pruned), autophagy regulates new protein synthesis, protein degradation and neurite elimination (axon or dendrite pruning) that is important to the formation of specific neuronal structures (Xilouri and Stefanis., 2010; Lee., 2012; Nixon and Cataldo., 2006). Thirdly, the elimination and establishment of synapses (synapse plasticity, and neuronal activity) are highly dynamically regulated in response to activity-dependent plasticity or Hebb’s synaptic plasticity that involves cytoskeletal dynamic mechanisms and multiple intracellular signaling cascades. This process needs a lot of molecules and /or proteins (such as neurotransmitters, RNA, and lipids) that are synthesized, delivered, and degraded for their dynamic functions in synaptic growth and synaptic activity. These are also regulated by autophagy (There is an introduction at the back of the manuscript: neuronal autophagy and synapse plasticity) (Hernandez et al., 2012).
Thus, we can say that autophagy is critical to maintaining the specialized functions of the neuron.
In addition, neurons must function continuously for as long as possible in a starved state, which is exceedingly different from other cell types. Brain neurons perform the functions of receiving, transmitting, integrating, processing, and outputting information, actively managing the general and cognitive activities of the organism. These processes place high energy and metabolic demand on
neurons, especially in axons and dendrites located far away from the soma. Previous studies suggested that neurons can use several different energy sources optimally, receive additional nutrients and neurotrophin support from glial cells, and benefit from hypothalamic regulation of peripheral nutrient supplies (Boland and Nixon., 2006). Autophagy can also regulate nutrient supply under even extreme starvation or diabetes. Notably, the activity-dependent refinement of synapses (including the neuronal circuit change)
occurs during the development and learning stages, and shares feature with diseases such as epilepsy.
For instance, the structural remodeling of mossy fiber synapses and the formation of aberrant synaptic contacts (such as synapse plasticity) in the dentate gyrus are common features in experimental models of epilepsy and human TLE (Cronin and Dudek., 1988; Sutula et al., 1988; Babb et al., 1991; Mello et al., 1993). In addition, amino acid metabolism (such as GABA) may be critical for epilepsy due to its role as the neurotransmitter (synapse transmission) and energy source (Bejarano and Rodríguez-Navarro., 2015). Epileptic seizures increase the energy requirements of the metabolically already compromised neurons, establishing a vicious cycle that results in worsening
energy failure and neuronal death (Desguerre et al., 2014). Changes in action potentials and ion channels are also related to the epileptogenesis of both genetic and acquired epilepsies (Armijo et al., 2005). Thus, changes in these physiological states may lead to the occurrence and development of epilepsy pathology.
References:
Xilouri M, Stefanis L. Autophagy in the central nervous system: implications for neurodegenerative disorders. CNS Neurol. Disord. Drug Targets. 2010; 9: 701-719.
Lee JA. Neuronal autophagy: a housekeeper or a fighter in neuronal cell survival? Exp Neurobiol. 2012; 21(1):1-8.
Nixon RA, Cataldo AM. Lysosomal system pathways: genes to neurodegeneration in Alzheimer's disease. J. Alzheimers Dis. 2006; 9:277-289.
Hernandez D, Torres CA, Setlik W, Cebrian C, Mosharov EV, Tang G, Cheng HC, Kholodilov N, Yarygina O, Burke RE, Gershon M, Sulzer D. Regulation of presynaptic neurotransmission by macroautophagy. Neuron. 2012; 74:277-284.
Boland B, Nixon RA. Neuronal macroautophagy: from development to degeneration. Mol Aspects Med. 2006; 27(5-6): 503-19.
Cronin J, Dudek FE. Chronic seizures and collateral sprouting of dentate mossy fibers after kainic acid treatment in rats. Brain Res 1988; 474: 181-4.
Sutula T, He XX, Cavazos J, et al. Synaptic reorganization in the hippocampus induced by abnormal functional activity. Science 1988; 239: 1147-50.
Babb TL, Kupfer WR, Pretorius JK, et al. Synaptic reorganization by mossy fibers in human epileptic fascia dentata. Neuroscience 1991; 42: 351-63.
Mello LE, Cavalheiro EA, Tan AM, et al. Circuit mechanisms of seizures in the pilocarpine model of chronic epilepsy: cell loss and mossy fiber sprouting. Epilepsia 1993;34: 985-95.
Bejarano E, Rodríguez-Navarro JA. Autophagy and amino acid metabolism in the brain: implications for epilepsy. Amino Acids. 2015; 47(10):2113-26.
Desguerre I, Hully M, Rio M, Nabbout R. Mitochondrial disorders and epilepsy. Rev Neurol (Paris). 2014; 170(5):375-80.
Armijo JA, Shushtarian M, Valdizan EM, Cuadrado A, de las Cuevas I, Adín J. Ion channels and epilepsy. Curr Pharm Des. 2005; 11(15):1975-2003.
Point 6: Line 417: “This apparent paradox may suggest that different types of neurons have unique axon architectures, arrangements, orientations, and different neuron types have dissimilar autophagy needs during axon regulation.” This statement at the end of a paragraph is not well supported by the evidence described earlier. What type of neurons show autophagy impairment in
axons?
Response 6: Thank you for your professional and valuable advice. This paradox is that abnormal autophagy induced by positive regulators exhibits different results in axon growth. Some cause neuroaxonal dystrophy and hypoplasia, while others promote axonal growth.
For instance, AP-4 is a member of the heterotetrameric adaptor protein (AP) family involved in protein sorting in the endomembrane system of eukaryotic cells (including neurons) (Hirst et al., 2013). δ2 glutamate receptor (δ2R) and amino-3-hydroxy-5-methyl-4-isoxazolepropionic acid (AMPA)-type glutamate receptors (AMPAR) are also sorted by AP-4 as cargos (Yap et al., 2003; Matsuda et al., 2008). The AP-6 4 ε (the ε subunit of AP-4) KO mice displays a thin corpus callosum and axonal swellings in various brain regions and spinal cord, such as hippocampal and Purkinje neurons, with a defect in autophagosome maturation. In addition, the AP-4 ε KO mice also displays impaired performance on the rotarod, and other motor and behavioral abnormalities, such as reduced grip strength, hindlimb clasping, increased ambulation, and enhanced acoustic startle response (Matsuda et al., 2008; De Pace et al., 2018). The disruption of the gene encoding the β subunit of AP-4 can increase the accumulation of axonal autophagosomes that contain AMPAR and transmembrane AMPA receptor regulatory proteins (TARPs) in axons of cerebellar Purkinje cells and hippocampal neurons both in vivo and in vitro.
In fact, AP-4 is associated with TARPs that tightly bind to all subunits of AMPA receptors GluR1-GluR4, whereby the mutation of AP-4 may regulate the proper somatodendritic-specific distribution of AMPA receptor, TARP, and other cargo in different neurons (such as the hippocampus, spinal cord, and other brain regions) (De Pace et al., 2018; Davies et al., 2018).
Moreover, knockouts of Atg5 and Atg7 in Purkinje neurons and rhodopsin neurons of the retina, and Atg7 in agouti-related peptide (AgRP) neurons of the hypothalamus have also been well-studied (Komatsu et al., 2006; Nishiyama et al., 2007; Kaushik et al., 2011; Chen et al., 2013; Zhou et al., 2015).
For instance, the disruption of the genes Atg5 and Atg7 caused Purkinje axonal swellings, followed by progressive dystrophy and degeneration in the axon (Komatsu et al., 2006; Nishiyama et al., 2007). Thus, the above studies suggest that autophagy deficiency induced by impairing positive regulators Ap4 (or ATG5, or ATG7) contributes to neuroaxonal dystrophy, axonal degeneration, and abnormal neuron functions, such as AMPAR-mediated synapse efficiency, particularly in Purkinje and hippocampal neurons. However, Ban and colleagues reported that the inhibition of autophagy by atg7 small interfering RNA (siRNA) resulted in the elongation of axons, while activation of
autophagy by rapamycin suppressed axon growth in cortical neurons (Ban et al., 2013).
Consistent with this, the mutation of Mir505-3p, which is an autophagic regulator for axonal elongation and branching, consistently enhanced the autophagy signaling cascade for axon growth and autophagosome biogenesis (Yang et al., 2017). So why does knocking out a positive regulator of
autophagy has the opposite effect? We note that the mammalian central nervous system needs autophagy to maintain its normal functions and homeostasis, particularly in the axons (Nixon., 2005; Xilouri and Stefanis., 2010; Lee., 2012; Hernandez et al., 2012). Different brain regions have distinct
types of neurons, synapse connections, and biological functions, which depend on dissimilar autophagy needs in different neurons (Xilouri and Stefanis., 2010; Hernandez et al., 2012). The deletion of positive regulators (such as AP-4, ATG5, and ATG7) displays axonal degeneration in different neurons of brain regions, particularly in hippocampal neurons, whereas the deletion of
positive regulators (such as ATG7) in cortical neurons displays axon growth. Thus, this paradox may suggest that different types of neurons (such as hippocampal, Purkinje, hypothalamic, and rhodopsin neurons) have different structures for specific neuron functions, and different neuron types have
dissimilar autophagy needs during axon regulation, which results in different effects for deletion of positive regulators.
References:
Hirst J, Irving C, Borner GH. Adaptor protein complexes AP-4 and AP-5: new players in endosomal trafficking and progressive spastic paraplegia. Traffic. 2013; 14(2):153-64.
Yap CC, Murate M, Kishigami S, Muto Y, Kishida H, Hashikawa T, Yano R. Adaptor protein complex- 4 (AP-4) is expressed in the central nervous system neurons and interacts with glutamate receptor delta2. Mol Cell Neurosci. 2003; 24(2):283-95.
Matsuda S, Miura E, Matsuda K, Kakegawa W, Kohda K, Watanabe M, Yuzaki M. Accumulation of AMPA receptors in autophagosomes in neuronal axons lacking adaptor protein AP-4. Neuron. 2008; 57(5):730-45.
De Pace R, Skirzewski M, Damme M, Mattera R, Mercurio J, Foster AM, Cuitino L, Jarnik M, Hoffmann V, Morris HD, Han TU, Mancini GMS, Buonanno A, Bonifacino JS. Altered distribution of ATG9A and accumulation of axonal aggregates in neurons from a mouse model of AP-4 deficiency syndrome. PLoS Genet. 2018; 14(4):e1007363.
Davies AK, Itzhak DN, Edgar JR, Archuleta TL, Hirst J, Jackson LP, Robinson MS, Borner GHH. AP-4 vesicles contribute to spatial control of autophagy via RUSC-dependent peripheral delivery of ATG9A. Nat Commun. 2018; 9(1):3958.
Ban BK, Jun MH, Ryu HH, Jang DJ, Ahmad ST, Lee JA. Autophagy negatively regulates early axon growth in cortical neurons. Mol Cell Biol. 2013; 33(19):3907-19.
Komatsu M, Waguri S, Chiba T, Murata S, Iwata J, Tanida I, Ueno T, Koike M, Uchiyama Y, Kominami E, Tanaka K. Loss of autophagy in the central nervous system causes neurodegeneration in mice. Nature. 2006; 441(7095):880-4.
Nishiyama J, Miura E, Mizushima N, Watanabe M, Yuzaki M. Aberrant membranes and double-membrane structures accumulate in the axons of Atg5-null Purkinje cells before neuronal death. Autophagy. 2007; 3(6):591-6.
Kaushik S, Rodriguez-Navarro JA, Arias E, Kiffin R, Sahu S, Schwartz GJ, Cuervo AM, Singh R. Autophagy in hypothalamic AgRP neurons regulates food intake and energy balance. Cell Metab. 2011; 14(2):173-83.
Chen Y, Sawada O, Kohno H, Le YZ, Subauste C, Maeda T, Maeda A. Autophagy protects the retina from light-induced degeneration. J Biol Chem. 2013; 288(11):7506-7518.
Zhou Z, Doggett TA, Sene A, Apte RS, Ferguson TA. Autophagy supports survival and phototransduction protein levels in rod photoreceptors. Cell Death Differ. 2015; 22(3):488-98.
Yang K, Yu B, Cheng C, Cheng T, Yuan B, Li K, Xiao J, Qiu Z, Zhou Y. Mir505-3p regulates axonal development via inhibiting the autophagy pathway by targeting Atg12. Autophagy. 2017; 13(10):1679-1696.
Lee JA. Neuronal autophagy: a housekeeper or a fighter in neuronal cell survival? Exp. Neurobiol. 2012; 21:1-8.
Nixon RA. Endosome function and dysfunction in Alzheimer's disease and other neurodegenerative diseases. Neurobiol. Aging. 2005; 26:373-382.
Xilouri M, Stefanis L. Autophagy in the central nervous system: implications for neurodegenerative disorders. CNS Neurol. Disord. Drug Targets. 2010; 9:701-719.
Hernandez D, Torres CA, Setlik W, Cebrian C, Mosharov EV, Tang G, Cheng HC, Kholodilov N, Yarygina O, Burke RE, Gershon M, Sulzer D. Regulation of presynaptic neurotransmission by macroautophagy. Neuron. 2012; 74:277-284.
Point 7: Some statements do not have references, for example in lines 649-662.
Response 7: Thank you for your kind suggestion. We have addressed this issue in the revised manuscript (in lines 775-789).
Point 8: The first paragraph of the section 6.2 may fit better at the end of the section 6.1.
Response 8: Thank you for your professional suggestion. This idea is awesome. We have put it at the end of Section 6.1 (In lines 690-705).
Point 9: The term “autophagy flux” is not well introduced.
Response 9: Thank you for your valuable advice. We added this point in this revised manuscript and a detailed revision can be found on lines 417-433.
Point 10: Line 624: “Consistently, research on epilepsy animal models also shows the increased ratio of LC3 II to LC3 I and decreased p62 protein levels after epileptic seizure onset, reflecting a blockage of autophagic flux...” This is not interpreted as a blockage of autophagy flux, but as a stimulation of autophagy flux. In addition, LC3-II/actin is a more accurate parameter than LC3-II/LC3-I ratio, as the detection of LC3-I is highly variable.
Response 10: Thank you very much for your professional introduction to LC3-II detection indicators in autophagy, we have benefited a lot. We have corrected this description in the latest manuscript (In line 751).
Point 11: Including a table listing the studies on neuronal autophagy in animal models and in human samples will help better to understand the scientific progress in this field.
Response 11: Thank you for your valuable suggestion. That is a good idea. We have added a table for understanding the scientific progress of autophagy and epilepsy (in lines 791-792).
Point 12: The Conclusion section does not clearly states what is known on neuronal autophagy and epilepsy.
Response 12: Thank you for your kind advice. We have added a clear description of epilepsy and autophagy in the conclusion section (in lines 809-820).
Dear Reviewers,
On behalf of all the contributing authors, I would like to express our sincere appreciation for your constructive comments concerning our article entitled “Molecular mechanism and regulation of autophagy and its potential role in epilepsy” (Manuscript ID: cells-1850560).
These comments are all valuable and helpful for improving our article. According to your professional and valuable comments, we have made extensive modifications to our manuscript to make our results convincing. In this revised version, changes to our manuscript were all highlighted within the document by using the “Track Changes” function of MS Word. Point-by-point responses to two reviewers are listed below this letter (Please see the attachment).
Thank you again for your positive comments and valuable suggestions to improve the quality of our manuscript. We have tried our best to improve and made some changes to the manuscript.
Best regards
Sincerely yours
Han-xiao Zhu,
The First Affiliated Hospital of Dali University,
32 Carlsberg Avenue, Dali City 671000, Yunnan, China,
e-mail: zhu_hanxiao1109@163.com

Round 2
Reviewer 2 Report
Comments 4 and 6 were satisfactorily addressed. Authors may consider to include a part or a summary of these responses in the manuscript.
Line 751: "Consistently, research on epilepsy animal models also shows the increased ratio of LC3 II to LC3 I and decreased p62 protein levels after epileptic seizure onset, reflecting an abnormal change of autophagic flux in kainic acid (KA)-induced rats and mice [243-244]. What authors mean with "abnormal change"? Increase or decrease of autophagy flux? That should be specified in the text.
Author Response
Dear Reviewers,
We feel very grateful for your professional review work on our article entitled “Molecular mechanism and regulation of autophagy and its potential role in epilepsy” (Manuscript ID: cells-1850560).
These comments are all valuable and helpful for improving our article. We have carefully revised the related issue again based on the latest file uploaded on August 12th. This revision is highlighted within the document by using the “Track Changes” function of MS Word. At the same time, we have uploaded the file of the revised manuscript. Point-by-point responses to your suggestions are listed below this letter (Please see the attachment).
Our deepest gratitude goes to you for your careful work and thoughtful suggestions that have helped improve this paper substantially.
If there are any other modifications we could make, we would like very much to modify them and we really appreciate your help. Thank you very much for your help.
Best regards
Sincerely yours
Han-xiao Zhu,
School of Clinical Medicine, Dali University,
The First Affiliated Hospital of Dali University,
32 Carlsberg Avenue, Dali City 671000, Yunnan, China,
E-mail: zhu_hanxiao1109@163.com
Response to Reviewer 2 Comments
Point 1: Comments 4 and 6 were satisfactorily addressed. Authors may consider to include a part or a summary of these responses in the manuscript.
Response 1: Thank you very much for your valuable advice. This idea is great. We have added these (in lines 436-489 and 558-605) to the latest manuscript.
Point 2: Line 751: "Consistently, research on epilepsy animal models also shows the increased ratio of LC3 II to LC3 I and decreased p62 protein levels after epileptic seizure onset, reflecting an abnormal change of autophagic flux in kainic acid (KA)-induced rats and mice [243-244]. What authors mean with "abnormal change"? Increase or decrease of autophagy flux? That should be specified in the text. Response 2: We are extremely grateful to the reviewer for pointing out this problem. The “abnormal change” represents the increase of autophagy activity in the paper. We have addressed this point in the revised revision (in line 812).

Round 3
Reviewer 2 Report
Authors have addressed all the comments successfully.